**EMBO** *reports*

# sVEGFR1 up-regulation via EGR1 impairs vascular repair in SFTSV-induced hemorrhage

Na Jiang [ID] [1,8], Jing Wu [ID] [1,8], Yating He [ID] [1,8], Rui Zhang[2], Mengmeng Ji [ID] [3], Linjing Zhu[1], Shengwei Cui [ID] [3], Qiao You [ID] [1], Yurong Cai[3], Bingxin Liu [ID] [1], Ruining Lyu [ID] [1], Yuxin Chen[4], Jin Zhu [ID] [5✉] & Zhiwei Wu [ID] [1,6,7✉]

## Abstract

**Hemorrhage is a major pathological manifestation of certain viral infections, such as severe fever with thrombocytopenia syndrome (SFTS), Ebola, Crimean-Congo hemorrhagic fever and Dengue. SFTS is an emerging viral hemorrhagic fever caused by the SFTS virus (SFTSV). Hemorrhage and angiogenesis dysfunction are key manifestations of SFTSV infection but the underlying mechanisms remain unclear. Here, we demonstrate that SFTSV infection increases soluble vascular endothelial growth factor-receptor 1 (sVEGFR1) secretion from monocytes/macrophages. Increased sVEGFR1 in the serum of SFTS patients is positively correlated with disease severity. Moreover, we show that SFTSV induces sVEGFR1 upregulation via early growth response gene 1 (EGR1), of which VEGFR1 is a downstream target. Serum from SFTS patients containing high levels of sVEGFR1 inhibit angiogenesis, which can be reversed by removal of sVEGFR1. Treatment of SFTSV-infected animals with sVEGFR1 neutralizing antibodies improves angiogenesis and prevents blood vessel leaks in vivo. In conclusion, we show that SFTSV infection induces sVEGFR1 secretion through EGR1 upregulation, thereby contributing to hemorrhage.**

**Keywords** SFTSV; Viral Hemorrhagic Fever; Vascular Dysfunction; sVEGFR1; EGR1
**Subject Categories** Microbiology, Virology & Host Pathogen Interaction; Signal Transduction; Vascular Biology & Angiogenesis

## Introduction

Hemorrhage is a common clinical manifestation of many severe viral infections, which often results in deleterious organ damages and even death (Chao et al, 2019; Iannetta et al, 2019; Karimi et al, 2016; Paessler and Walker, 2013; Shimizu et al, 2017; Xu et al, 2021; Zhang et al, 2012). Hemorrhagic fever-inducing viruses tend to cause thrombocytopenia and severely impair the ability of blood clotting, thus leading to the hemorrhagic outcome in severe cases of infection, which is considered as a major pathophysiological manifestation that are highly associated with eventual death (Feldmann and Geisbert, 2011; Messaoudi and Basler, 2015). However, thrombocytopenia alone is not sufficient to lead to hemorrhage without the disruption of vascular endothelial integrity. For viruses that cause hemorrhagic fever, vascular endothelial leakage is not only an outcome of pathophysiological manifestation but also a way of viral transmission. How virus infection causes the disruption of vascular endothelial integrity and thus impacts on the hemorrhage are not fully understood.

Severe fever with thrombocytopenia syndrome (SFTS) is a viral infectious disease caused by SFTS virus (SFTSV), which has a significant impact on public health worldwide (Yu et al, 2011). Since its discovery in 2009, SFTS has been increasingly reported in the central, eastern, and northern regions of China and other Asian countries, including South Korea, Japan, Vietnam, and Myanmar with expanding spatial distribution (Denic et al, 2011; Kim et al, 2013; Liang et al, 2023; McMullan et al, 2012; Sun et al, 2017; Tran et al, 2019; Win et al, 2020; Zohaib et al, 2020). It is characterized by acute fever, leukopenia and thrombocytopenia with case fatality rates as high as 30% (Li et al, 2021b). Due to its high fatality rate, SFTS had been listed in the top 10 infectious diseases requiring prioritized research and intervention by WHO in 2017 (Wang et al, 2021). No licensed vaccines or specific antiviral therapies have yet been developed for the prevention and treatment of SFTS, thus raising concerns that SFTS will pose a severe public health threat.

The clinical manifestations of SFTS range from asymptomatic carriage, influenza-like illnesses, hemorrhagic fever to multiple organ failures and even death. The major pathophysiologic feature of severe SFTS is leukopenia and thrombocytopenia, which are generally the key clinical manifestations in severe cases. However, an acute increase in vascular permeability, leading to fluid leakage into tissues and severe hemorrhage were reported (Li et al, 2017), suggesting that the virus may disrupt the vascular protective function and affect subsequent vascular repair process, which were considered the most relevant event in severe and fatal SFTS pathogenesis (Zhang et al, 2012). There are two theories proposed

[1]Center for Public Health Research, Medical School, Nanjing University, Nanjing, China. [2]Department of Infectious Diseases, Nanjing Drum Tower Hospital, Affiliated Hospital of Medical School, Nanjing University, Nanjing, China. [3]School of Life Sciences, Ningxia University, Yinchuan, China. [4]Department of Laboratory Medicine, Nanjing Drum Tower Hospital, Affiliated Hospital of Medical School, Nanjing University, Nanjing, China. [5]Huadong Medical Institute of Biotechniques, Zhongshan Dong Road 293, Nanjing, China. [6]Yunnan Provincial Key Laboratory of Entomological Biopharmaceutical R&D, College of Pharmacy, Dali University, Dali, China. [7]State Key Laboratory of Analytical Chemistry for Life Science, Nanjing University, Nanjing, China. [8]These authors contributed equally: Na Jiang, Jing Wu, Yating He. ✉E-mail: zhujin1968@njmu.edu.cn; wzhw@nju.edu.cn

for the viral hemorrhagic fever pathogenesis: (1) the virus infects the endothelial cells directly, resulting in disruption of the vascular endothelium, and (2) the virus acts indirectly by infecting immune cells, such as monocytes/macrophages, the major target of SFTSV (Peng et al, 2016; Qu et al, 2012; Suzuki et al, 2020; Zhang et al, 2019), and inducing the release of soluble mediators (Akinci et al, 2013). As with many RNA viral infections, SFTSV infects monocytes/macrophages and uses these cells for dissemination, replication and long-term persistence within tissues. However, the roles of monocytes/macrophages in the potential pathogenic mechanisms of hemorrhage syndrome remain unclear.

Monocytes/macrophages, being the major cellular target for SFTSV infection serve as the source of the various cytokines upon infection (Zhang et al, 2019). Monocytes/macrophages produce high levels of soluble vascular endothelial growth factor-receptor 1 (sVEGFR1) and exhibit anti-angiogenic properties (Eubank et al, 2004). Monocytes/macrophages contribute to the inhibition of tumor angiogenesis, in part, through their secretion of the potent anti-angiogenic molecule, sVEGFR1 (Liu, 2023). sVEGFR1 is the most important factor that antagonizes the pro-angiogenic effects of vascular endothelial growth factor-A (VEGFA) (Melincovici et al, 2018; Shibuya, 2013).

In the current study, we demonstrate that SFTSV infection induces secretion of sVEGFR1 from monocytes/macrophages and that SFTSV infection inhibits angiogenesis in an in vivo model, which is reversed by the removal of sVEGFR1. Additionally, we show that sVEGFR1 upregulation is caused by SFTSV-induced early growth response gene-1 (EGR1) that acts on VEGFR1. Treatment with sVEGFR1 neutralizing antibodies ameliorates the inhibition of angiogenesis in an in vivo SFTSV infection model. Thus, our findings suggest that the EGR1/sVEGFR1/ hemorrhage axis might be a potential target for the development of novel anti-hemorrhagic therapies during SFTSV infection.

# Results

## SFTSV infection triggers the release of sVEGFR1 from monocytes/macrophages

Monocytes/macrophages are the major cellular target of SFTSV (Zhang et al, 2019) and the infection of monocytes/macrophages triggers a cascade of immunopathology, including pro-inflammatory immune response (Li et al, 2021a; Park et al, 2021). To explore the role of monocytes/macrophages in the hemorrhage syndrome and the potential pathogenic mechanisms, THP-1 cells, a human monocytic cell line which is susceptible to SFTSV, were used as an in vitro infection model of monocytes/macrophages (Bosshart and Heinzelmann, 2016; Zhang et al, 2019). We found that the mRNA level of sVEGFR1, which has a minimal basal level expression under normal condition, increased continuously with SFTSV infection and time as determined by qPCR using sVEGFR1-specific primers (Fig. 1A). Consistently, the protein level of sVEGFR1 also showed an increase with infection time (Fig. 1B). Since sVEGFR1 is minimally expressed in non-infected cells and an important negative modulator of angiogenesis by sequestering VEGFA and VEGFR2 activation in the circulatory system (Garvin et al, 2005), we investigated the secretion of sVEGFR1 during SFTSV infection. As shown in Fig. 1C, the sVEGFR1 in

supernatants of SFTSV-infected cells was increased compared to the uninfected cells, and the increase became significant as the infection progressed from 24 to 60 h post-infection.

To confirm this observation, THP-1 cells were infected with SFTSV at different multiplicity of infections (MOIs) and the mRNA level of sVEGFR1 was shown upregulated in a dose-dependent manner (Fig. 1D–H). Furthermore, we also observed increased sVEGFR1 secretion responding to increasing infectious dose (Fig. 1I–M). Collectively, these results unequivocally confirmed that SFTSV infection triggers the release of sVEGFR1 from THP-1 cells.

qPCR and flow cytometry analysis showed that total VEGFR1 mRNA and protein expression on the cell membrane were significantly increased in THP-1 cells (Fig. EV1A,B), suggesting that SFTSV infection upregulated de novo production of sVEGFR1. Thus, SFTSV-induced sVEGFR1 was not generated by cleaving membrane-bound VEGFR1 and instead was modulated at transcriptional level. As shown in Fig. EV1C,D, THP-1 cells were induced to differentiate into macrophage-like cells by PMA and SFTSV infection significantly increased the total VEGFR1 and sVEGFR1 mRNA.

We also conducted experiments using primary human monocyte-derived macrophages to validate our findings. The results showed that the mRNA levels of sVEGFR1 (Fig. 1N) and total VEGFR1 (Fig. EV1E) were increased in SFTSV-infected macrophages and the sVEGFR1 in supernatants of SFTSV-infected macrophages was increased compared to the uninfected macrophages (Fig. 1O). We also examined sVEGFR1 expression in BMDMs from C57BL/6J mice, and showed that SFTSV infection upregulated sVEGFR1 expression (Fig. 1P). These results demonstrate that SFTSV infection triggered the release of sVEGFR1 from primary macrophages.

In an earlier study, we showed that SFTSV infection drove macrophage differentiation skewed to M2 phenotype (Zhang et al, 2019). Consistently, when THP-1 cells were induced to differentiate into either M1 macrophage-like cells or M2 macrophage-like cells, we found that both total VEGFR1 and sVEGFR1 mRNA were significantly higher in M2-like cells than in M1-like cells (Fig. EV2A,B), suggesting that sVEGFR1 was mainly produced by M2 macrophage-like THP-1 cells. Furthermore, when primary human monocyte-derived macrophages were differentiated into either M1 macrophages or M2 macrophages, we found that the mRNA levels of total VEGFR1 and sVEGFR1 were increased in M2 macrophages (Fig. EV2C,D). To further assess the secretion of sVEGFR1 in the supernatants, we performed an ELISA and demonstrated that M2 macrophages produced significantly higher sVEGFR1 in the supernatants as compared with that of M1 macrophages (Fig. EV2E). Collectively, these results suggest that sVEGFR1 was mainly produced by M2 macrophages.

## SFTSV infection triggers the release of sVEGFR1 in humanized mice and SFTS patients

The effect of SFTSV infection on the expression of sVEGFR1 was then evaluated in hu-PBL NCG mice. This SFTSV infection mouse model was developed to present a number of pathogenic manifestations similar to human infection (Xu et al, 2021). sVEGFR1 mRNA in the blood was 8-fold higher in SFTSV-infected mice than uninfected mice (Fig. 2A). To determine the

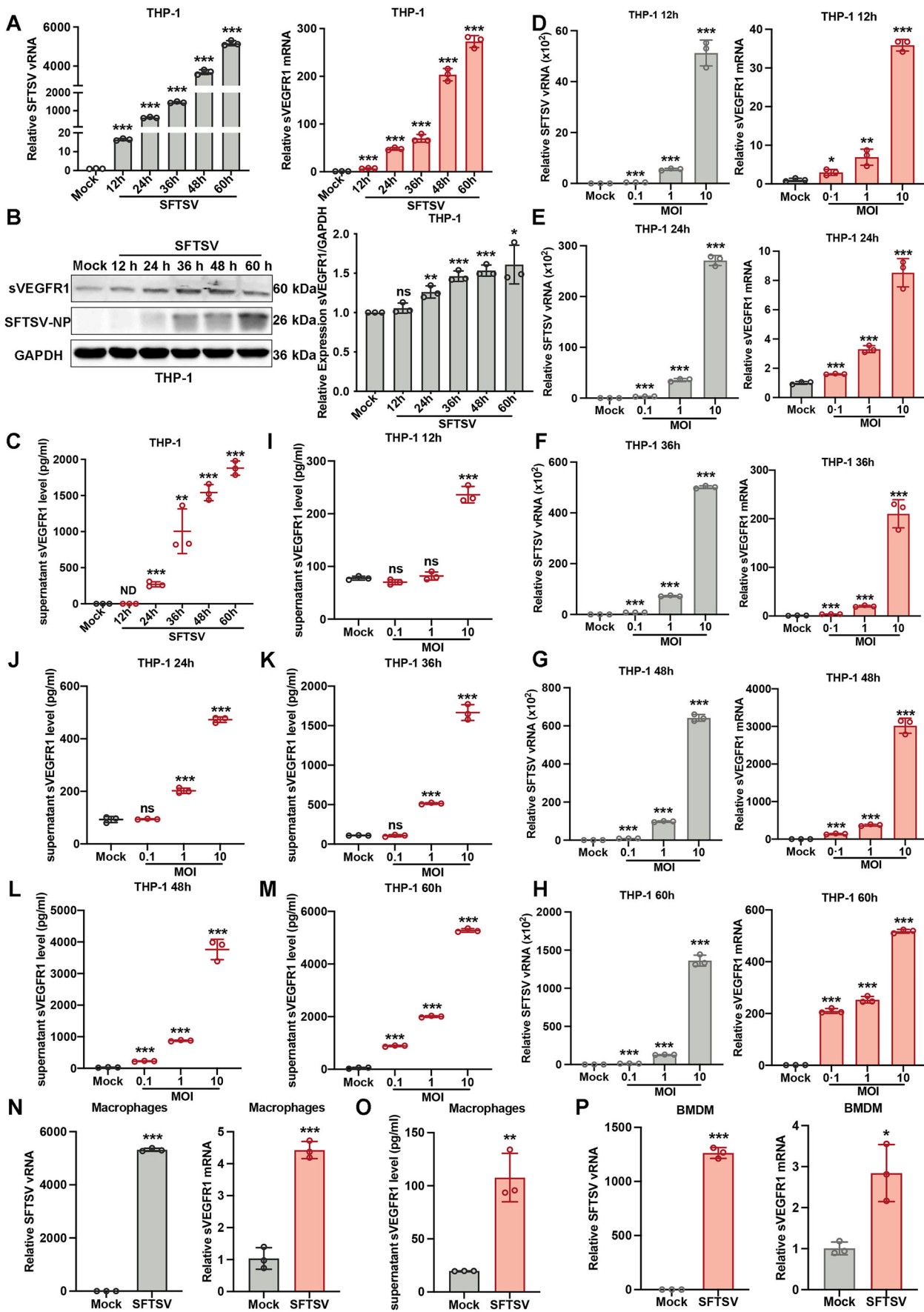

**Figure 1. SFTSV infection up-regulates sVEGFR1 expression in monocytes/macrophages.**

(A) THP-1 cells were infected with SFTSV (MOI = 1) for 12, 24, 36, 48, and 60 h. SFTSV viral RNA (left) and sVEGFR1 mRNA (right) were measured by qPCR. n = 3 biological replicates. Statistical significance was determined by two-tailed unpaired Student t test. (SFTSV: ***$P < 0.0001$, ***$P < 0.0001$, ***$P < 0.0001$, ***$P < 0.0001$, ***$P < 0.0001$; sVEGFR1: ***$P = 0.0002$, ***$P < 0.0001$, ***$P = 0.0001$, ***$P < 0.0001$, ***$P < 0.0001$). (B) THP-1 cells were infected with SFTSV for 12, 24, 36, 48, and 60 h. Intracellular sVEGFR1 and the viral NP protein were measured by western blotting. sVEGFR1 was stained with a specific monoclonal antibody and GAPDH served as an internal control. n = 3 biological replicates. Statistical significance was determined by two-tailed unpaired Student t test. (ns = 0.2035, **$P = 0.0043$, ***$P = 0.0003$, ***$P = 0.0002$, *$P = 0.0127$). (C) THP-1 cells were infected with SFTSV for 12, 24, 36, 48 and 60 h. The secreted sVEGFR1 in cell supernatant was measured by capture ELISA. n = 3 biological replicates. Statistical significance was determined by two-tailed unpaired Student t test. (ND: Not Detected, ***$P = 0.0002$, **$P = 0.0049$, ***$P < 0.0001$, ***$P < 0.0001$). (D–H) THP-1 cells were infected with SFTSV for 12, 24, 36, 48 and 60 h at various MOIs (MOI = 0.1, 1, and 10). Viral RNA (left) and sVEGFR1 mRNA (right) were determined by qPCR. n = 3 biological replicates. Statistical significance was determined by two-tailed unpaired Student t test. (12 h: left: ***$P < 0.0001$, ***$P < 0.0001$, ***$P < 0.0001$; right: *$P = 0.0169$, **$P = 0.0087$, ***$P < 0.0001$. 24 h: left: ***$P < 0.0001$, ***$P < 0.0001$, ***$P < 0.0001$; right: ***$P = 0.0005$, ***$P < 0.0001$, ***$P = 0.0002$. 36 h: left: ***$P < 0.0001$, ***$P < 0.0001$, ***$P < 0.0001$; right: ***$P = 0.0004$, ***$P < 0.0001$, ***$P = 0.0002$. 48 h: left: ***$P < 0.0001$, ***$P < 0.0001$, ***$P < 0.0001$; right: ***$P < 0.0001$, ***$P < 0.0001$, ***$P < 0.0001$. 60 h: left: ***$P < 0.0001$, ***$P < 0.0001$, ***$P < 0.0001$; right: ***$P < 0.0001$, ***$P < 0.0001$, ***$P < 0.0001$). (I–M) THP-1 cells were infected with SFTSV for 12, 24, 36, 48 and 60 h at various MOIs (MOI = 0.1, 1, and 10). sVEGFR1 in cell supernatant was measured by capture ELISA. n = 3 biological replicates. Statistical significance was determined by two-tailed unpaired Student t test. (12 h: ns = 0.0910, ns = 0.4550, ***$P < 0.0001$; 24 h: ns = 0.8551, ***$P = 0.0002$, ***$P < 0.0001$; 36 h: ns = 0.6919, ***$P < 0.0001$, ***$P < 0.0001$; 48 h: ***$P < 0.0001$, ***$P < 0.0001$, ***$P < 0.0001$; 60 h: ***$P < 0.0001$, ***$P < 0.0001$, ***$P < 0.0001$). (N) Primary human monocyte-derived-macrophages were infected with SFTSV (MOI = 1) for 24 h. SFTSV viral RNA (left) and sVEGFR1 mRNA (right) were measured by qPCR. n = 3 biological replicates. Statistical significance was determined by two-tailed unpaired Student t test. (SFTSV: ***$P < 0.0001$; sVEGFR1: ***$P = 0.0002$). (O) Primary human monocyte-derived-macrophages were infected with SFTSV (MOI = 1) for 24 h. The secreted sVEGFR1 in cell supernatant was measured by capture ELISA. n = 3 biological replicates. Statistical significance was determined by two-tailed unpaired Student t test. (**$P = 0.0026$). (P) Bone marrow-derived macrophages (BMDMs) from C57BL/6J mice were matured to macrophage and then infected with SFTSV (MOI = 1) for 24 h. BMDMs were infected with SFTSV for 24 h. Viral RNA (left) and sVEGFR1 (right) mRNA were measured by qPCR. n = 3 biological replicates. Statistical significance was determined by two-tailed unpaired Student t test. (SFTSV: ***$P < 0.0001$; sVEGFR1: *$P = 0.0111$). Data information: Data shown are mean ± SD of three biological replicates. (ns, $P > 0.05$; *$P < 0.05$; **$P < 0.01$; ***$P < 0.001$). Source data are available online for this figure.

capability of SFTSV inducing sVEGFR1 secretion, we performed an ELISA and demonstrated that SFTSV-infected mice produced significantly higher sVEGFR1 in the serum as compared with that of the uninfected animals by up to 15-fold (Fig. 2B). At the same time, the protein level of sVEGFR1 in spleens were increased after SFTSV infection (Fig. 2C). Consistently, immunofluorescence analysis showed that sVEGFR1 increased robustly in the spleens of SFTSV-infected mice (Fig. 2D). Collectively, these results demonstrated that SFTSV infection triggered sVEGFR1 expression and secretion in hu-PBL NCG mice.

We further demonstrated that blood sVEGFR1 level was significantly elevated in SFTS patients. There was statistically significant difference for sVEGFR1 between the healthy individuals and the patients (Fig. 2E). sVEGFR1 levels were higher in SFTS patients than the healthy individuals and the severe SFTS patients had significantly higher blood sVEGFR1 than the patients with mild symptoms ($P < 0.0001$), suggesting a strong pathophysiological correlation between sVEGFR1 and the disease severity. Therefore, sVEGFR1 may be an important biomarker for the risk of severity and death as a result of infection with SFTSV.

## SFTSV-induced sVEGFR1 secretion increases angiogenic dysfunction in vitro

Hemorrhage is a hallmark of SFTS (Li et al, 2018a); we, therefore, hypothesized that sVEGFR1 was involved in the angiogenic dysfunction induced by SFTSV infection. To demonstrate the role of SFTSV-induced sVEGFR1 in angiogenesis, we performed a tube formation assay on HUVECs using sera collected from SFTS patients of various clinical severity. Severe SFTS patients were defined by the presence of hemorrhagic manifestations, any one or more organ failure or encephalitis development (Lu et al, 2015). As shown in Fig. 3A, the tube formation of HUVECs was significantly inhibited in these groups treated with sera of mild and severe infections compared with the healthy group. The aortic ring assay was also performed to determine the effect of the sera of SFTS

patients on angiogenesis. As shown in Fig. 3B, treatment of aortic rings with the sera collected from SFTS patients significantly inhibited capillary outgrowth. Furthermore, when sVEGFR1 was removed from the patients' sera using sVEGFR1 neutralizing antibody-conjugated magnetic beads, as indicated in Fig. 3C, the tube formation of HUVECs showed a significant enhancement (Fig. 3C, lower right panel). Treatment of HUVECs with sera from healthy individuals supported normal tube formation, which was significantly reduced by treatment with recombinant sVEGFR1 protein (Fig. 3D). Conversely, the treatment of HUVECs with sera from SFTS patients markedly impaired tube formation, which could be partially restored by treatment with recombinant VEGFA protein. These results suggested that SFTSV-induced sVEGFR1 specifically contributed to angiogenic dysfunction.

## SFTSV infection up-regulates EGR1

To determine the mechanism underlying the inducible expression of sVEGFR1, we performed a transcriptome analysis in SFTSV-infected or mock-infected HUVECs. The heatmap of RNA-seq showed that 81 genes relating to angiogenesis were among 422 genes with significant differences in expression after SFTSV infection (Appendix Table S1) and the early growth response 1 (EGR1) was strongly expressed in SFTSV-infected cells (Fig. 4A,B). To investigate whether EGR1 was modulated by SFTSV infection, we verified that SFTSV infection promoted EGR1 protein expression by twofold in both HeLa (Fig. 4C) and THP-1 cells (Fig. 4D). In addition, the mRNA levels of EGR1 were also increased in SFTSV-infected primary human monocyte-derived-macrophages compared to uninfected macrophages (Fig. 4E). We also examined EGR1 expression in BMDMs from C57BL/6J mice, and showed that SFTSV infection upregulated EGR1 expression (Fig. 4F). Next, we investigated whether SFTSV infection could induce EGR1 expression in vivo. hu-PBL NCG mice were inoculated with $1 \times 10^3$ TCID$_{50}$ of SFTSV, and the whole blood and the spleen were collected for analysis of EGR1 expression.

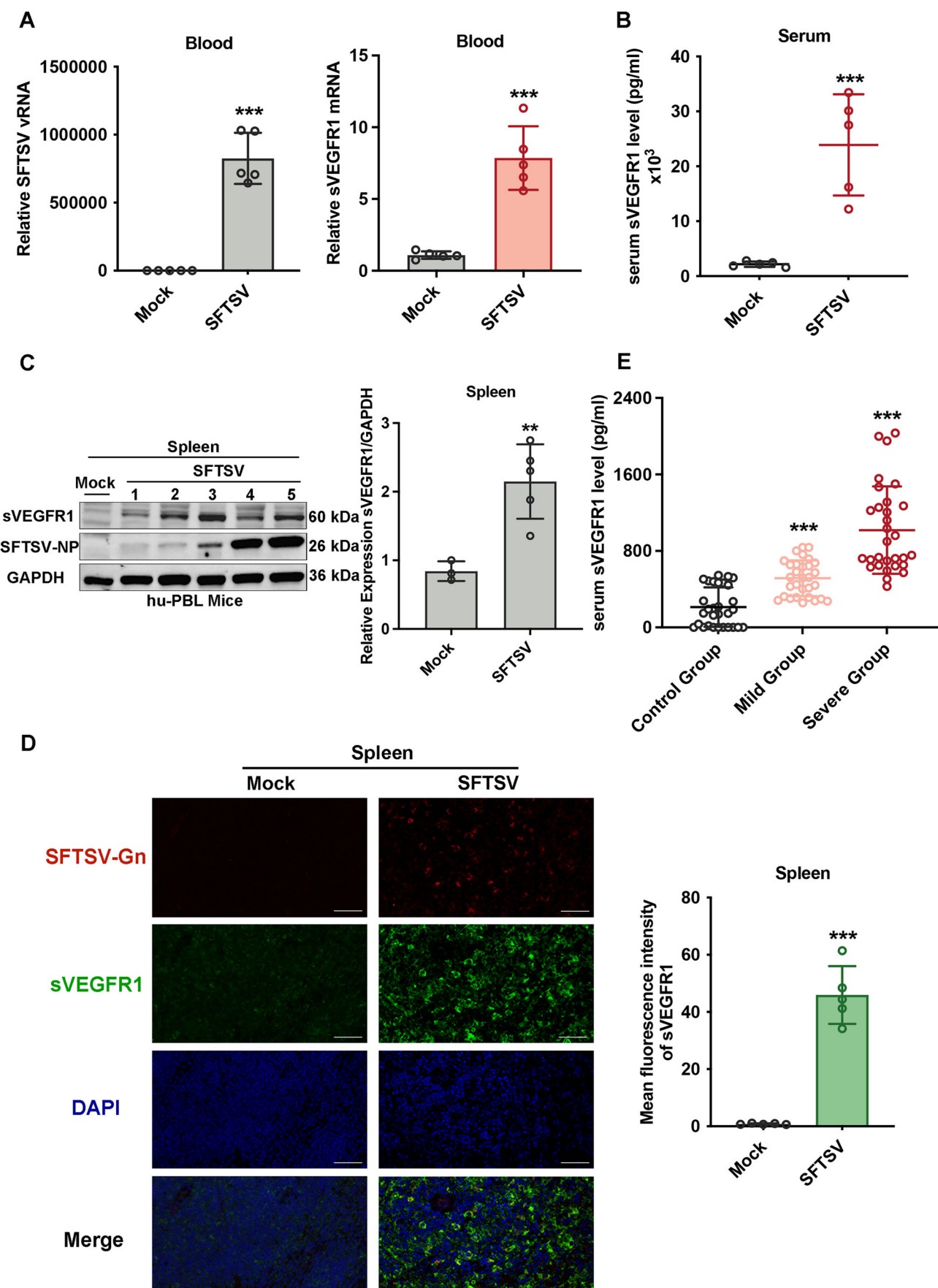

**Figure 2. SFTSV infection up-regulates sVEGFR1 expression in hu-PBL NCG mice and in SFTS patients.**

(A) hu-PBL NCG mice were infected with SFTSV ($n = 5$) at $1 \times 10^3$ TCID$_{50}$ or mock-infected ($n = 5$), and blood samples were collected after 14 days. SFTSV viral RNA (left) and sVEGFR1 mRNA (right) in blood cells were determined by qPCR. $n = 5$ biological replicates. Statistical significance was determined by two-tailed unpaired Student $t$ test. (SFTSV, ***$P < 0.0001$; sVEGFR1, ***$P = 0.0001$). (B) The serum level of sVEGFR1 protein was measured by capture ELISA. Each point represents an individual. $n = 5$ biological replicates. Statistical significance was determined by two-tailed unpaired Student $t$ test. (***$P = 0.0008$). (C) The infected mice ($n = 5$) were euthanized, and spleens were collected. sVEGFR1 and the viral NP protein in spleens were measured by western blotting (left). Numeric label indicates individual mice. The sVEGFR1 protein expression relative to GAPDH was shown (right). $n = 3$ biological replicates for Mock; $n = 5$ biological replicates for SFTSV. Statistical significance was determined by two-tailed unpaired Student $t$ test. (**$P = 0.0074$). (D) Detection of sVEGFR1 by immunofluorescence in the spleens after SFTSV infection (left). SFTSV-Gn (red), sVEGFR1 (green) and nuclei (DAPI, blue) were visualized using either antibodies specific for Gn or sVEGFR1, or DAPI for nuclei. A merged image of red, green and blue channels is shown (merge). The quantifications of relative sVEGFR1 intensities were determined by ImageJ software (right). Scale bars, 25 μm, respectively. $n = 5$ biological replicates. Statistical significance was determined by two-tailed unpaired Student $t$ test. (***$P < 0.0001$). (E) Blood samples were obtained from 60 SFTS patients (grouped into severe and mild based upon clinical criteria) and 30 healthy individuals as controls. The serum level of sVEGFR1 was measured by capture ELISA. $n = 30$ biological replicates for each group. Statistical significance was determined by two-tailed unpaired Student $t$ test. (Mild, ***$P < 0.0001$; Severe, ***$P < 0.0001$). Data information: Data shown are mean ± SD of at least three biological replicates with each data point representing a biological experiment (**$P < 0.01$; ***$P < 0.001$). Source data are available online for this figure.

EGR1 mRNA in blood was 10-fold higher in SFTSV-infected mice than uninfected mice (Fig. 4G) while the expression of EGR1 protein in spleen was also upregulated in response to SFTSV infection (Fig. 4H). Consistently, we observed by immunohisto-chemical staining that EGR1 increased in the spleen of SFTSV-infected mice (Fig. 4I). In addition, we found that the mRNA level of EGR1 was increased in M2 macrophages (Fig. EV2F) compared to M1 macrophages. EGR1 expression in PBMCs among SFTS patients were measured by qPCR and the results showed that the mRNA level of EGR1 were increased in PBMCs from infected patients (Fig. EV3). Collectively, these results demonstrated that SFTSV infection upregulated EGR1 both in vitro and in vivo.

## EGR1 positively regulates the expression of sVEGFR1

sVEGFR1 contains regulatory element for EGR1 and is regulated by EGR1 (Jin et al, 2009; Vidal et al, 2000). To explore the effect of increased EGR1 expression on sVEGFR1 induction, HeLa cells were transiently transfected with a plasmid expressing EGR1 (EGR1-his) or an empty vector (EV-his). We observed that the overexpression of EGR1 upregulated sVEGFR1 at both the mRNA (Fig. 5A) and protein (Fig. 5B) levels. Subsequently, sVEGFR1 in cell supernatant was also increased significantly by EGR1 over-expression (Fig. 5C). We also established a HEK-293T cell line stably overexpressing EGR1 (EGR1-293T cells) and showed that EGR1 and sVEGFR1 were both significantly increased in EGR1-293T cells at the mRNA (Fig. 5D) and protein levels (Fig. 5E), respectively. We also found that the level of sVEGFR1 in EGR1-293T cell cultural supernatant was elevated significantly compared with the control cell supernatant (Fig. 5F). Previous studies demonstrated that VEGFR1 promoter sequence contains a binding site for EGR1 at positions -24 to -16, and EGR1 was able to bind to this DNA sequence, resulting in enhanced transcriptional activity (Vidal et al, 2000). As shown in Fig. EV4, the results indicated that EGR1 could enhance the luciferase activity of VEGFR1-promoter.

To further examine the effects of EGR1 on sVEGFR1 expression, knock-down of EGR1 with siRNA in HEK-293T cells by transfecting the cells with EGR1 siRNA significantly reduced the sVEGFR1 protein expression as shown by western blotting (Fig. 5G). In addition, knock-down of EGR1 with siRNA in THP-1 cells significantly reduced the sVEGFR1 expression as shown by qPCR (Fig. EV5), further demonstrating the important role of EGR1 in promoting sVEGFR1 expression.

## Blocking sVEGFR1 improves angiogenesis in vivo during SFTSV infection

To further confirm the antagonistic role of sVEGFR1 in angiogenesis during SFTSV infection, a Matrigel plug angiogenesis assay was performed in vivo (Fig. 6A). As shown in Fig. 6B, Matrigel plugs containing a control antibody in SFTSV-infected mice exhibited white color, suggesting minimal blood vessel formation. Plugs containing a sVEGFR1 neutralizing antibody in SFTSV-infected mice presented as light red color, indicating that new blood vessel formation in the plugs. H&E staining showed that the density of blood vessels was significantly higher in the plugs containing a sVEGFR1 neutralizing antibody as compared to those in the control plugs (Fig. 6C). Immunohistochemical analysis further revealed the increased expression of endothelial cell marker CD31 in the Matrigel plug containing sVEGFR1-specific neutralizing Ab in both infected and uninfected mice (Fig. 6D). These results indicated that blocking sVEGFR1 by a neutralizing antibody restored angiogenesis in vivo during SFTSV infection.

## sVEGFR1 expression is downregulated in EGR1 KO mice during SFTSV infection, alleviating angiogenesis dysfunction

To further confirm the role of EGR1 in hemorrhage during SFTSV infection, we carried out SFTSV infection in EGR1 knockout (EGR1$^{-/-}$) mice. The efficacy of EGR1 knockout was confirmed by PCR and qPCR. EGR1 was not detected in EGR1$^{-/-}$ mice (Fig. 7A), confirming the deletion of EGR1 gene. The adult immunocompetent mouse model can only partially replicate the characteristics of human SFTSV infections, as mice do not develop severe clinical manifestations or succumb to SFTSV infection. Among existing immunocompromised mouse models, mice pre-treated with an IFNAR-specific antibody can develop lethal symptoms and exhibit severe clinical manifestations, including hemorrhage, similar to those observed in humans. So EGR1 KO mice were pre-injected with IFNAR1 Ab and the physical states of the IFNAR Ab-treated mice were monitored for 7 days after the SFTSV challenge (Fig. 7B). As shown in Fig. 7C,D, the WT mice experienced symptom aggravation until death and all died within 6 days, whereas 80% of the KO mice survived for 7 days, although they all exhibited significant weight loss. We next determined the expression of blood sVEGFR1 and showed that the levels of sVEGFR1 in the KO mice decreased significantly compared with the WT mice (Fig. 7E).

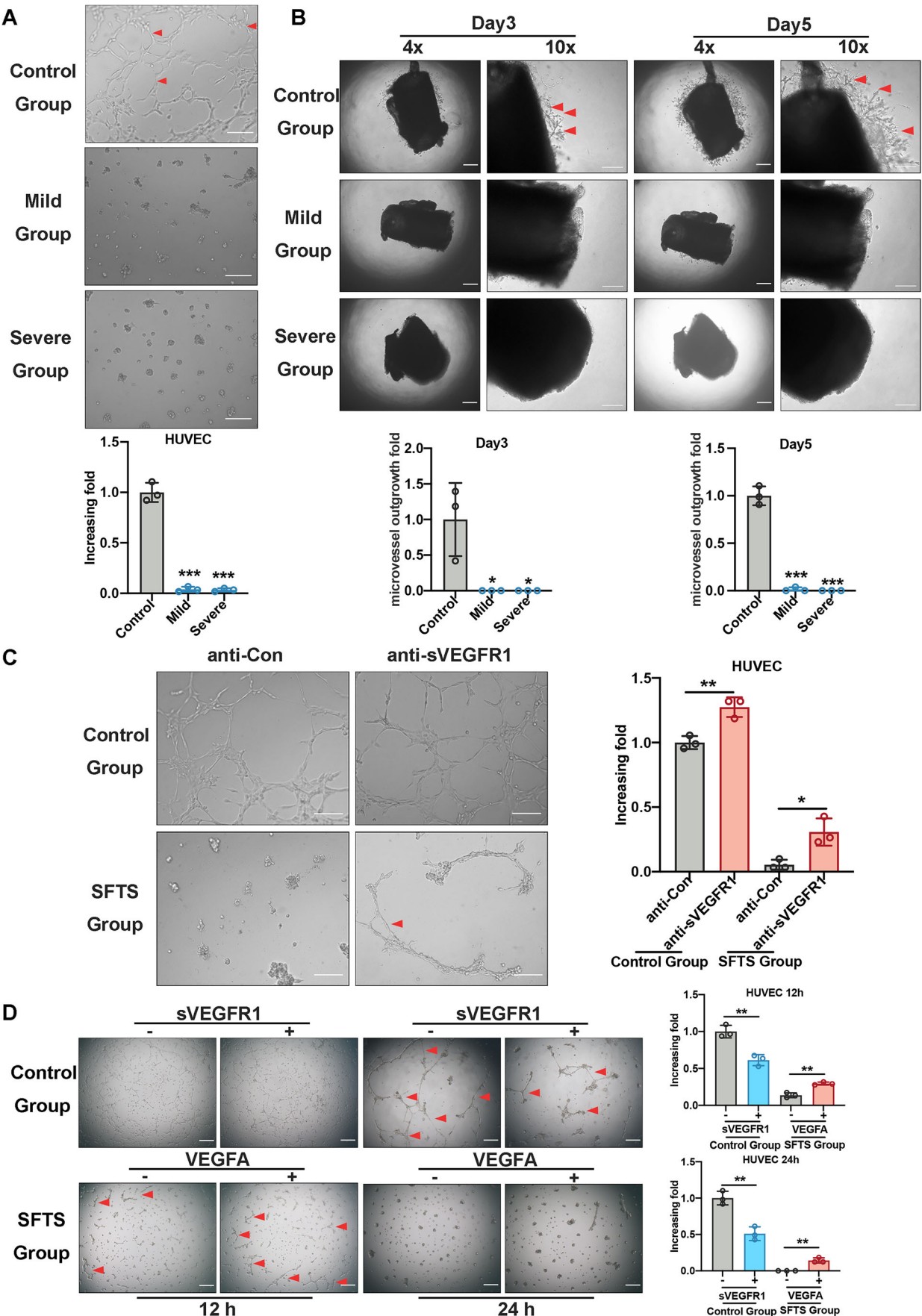

**Figure 3. SFTSV-upregulated sVEGFR1 induces angiogenesis dysfunction in vitro.**

(A) HUVECs were introduced into the Matrigel in the presence of serum. Each serum collected from SFTS patients of either mild symptom or severe symptom was placed under UV light for 30 min for residual virus inactivation. Tube formation of HUVECs was examined at 12 h after treatment. Representative capillary tubule structures were shown (top). Red arrowheads indicate tube elongation. Bar graph (bottom) represent the fold change of tubule formation. Scale bar = 50 μm. $n = 3$ biological replicates for each group. Statistical significance was determined by two-tailed unpaired Student $t$ test. (Mild, ***$P < 0.0001$; Severe, ***$P < 0.0001$). (B) Representative images (top) of mouse aortic rings from which microvessels sprouted after treatment with sera collected from either mild patients or severe patients. Each serum was placed under UV light for 30 min for residual virus inactivation. Red arrowheads indicate micro-vascular sprouts. Bar graph (bottom) represent the fold change of microvessels outgrowth from aortic rings. Scale bar = 50 μm. $n = 3$ biological replicates for each group. Statistical significance was determined by two-tailed unpaired Student $t$ test. (Day 3: *$P = 0.0281$, *$P = 0.0281$; Day 5: ***$P < 0.0001$, ***$P < 0.0001$). (C) Sera from healthy individuals and SFTS patients were placed under UV light for 30 min to inactivate the virus, and then pre-treated with a sVEGFR1 neutralizing antibody-conjugated or control antibody-conjugated magnetic beads to remove sVEGFR1. HUVECs were introduced into the Matrigel in the presence of pre-treated-serum from either healthy individuals or SFTS patients, and tube formation of HUVECs was examined at 12 h after treatment. Representative capillary tubule structures were shown. Red arrowheads indicate tube elongation. Scale bar = 50 μm. Bar graph on the right represent the fold change of tubule formation. $n = 3$ biological replicates for each group. Statistical significance was determined by two-tailed unpaired Student $t$ test. (Control, **$P = 0.0065$; SFTS, *$P = 0.0177$). (D) Representative images of the tube formation following incubation with pre-treated sera from either healthy individuals or SFTS patients added with exogenous recombinant sVEGFR1 (10 ng/ml) or VEGFA (20 ng/ml) protein, respectively. Red arrowheads indicate tube elongation. Scale bar = 50 μm. Bar graph on the right represent the fold change of tubule formation. $n = 3$ biological replicates for each group. Statistical significance was determined by two-tailed unpaired Student $t$ test. (12 h: **$P = 0.0041$, **$P = 0.0021$; 24 h: **$P = 0.0030$, **$P = 0.0022$). Data information: Data shown are mean ± SD of three biological replicates. (*$P < 0.05$; **$P < 0.01$; ***$P < 0.001$). Source data are available online for this figure.

There was a significant decline in the levels of viral RNA in the KO mice compared with the WT mice as determined by qPCR (Fig. 7E). We further performed an ELISA and demonstrated that SFTSV infected-KO mice produced significantly lower sVEGFR1 in the serum as compared with that of the infected-WT mice (Fig. 7F). We analyzed the pathological lesions in various tissues by H&E staining and pathological manifestations were identified in the livers, spleens, lungs, and kidneys (Fig. 7G). Histopathological analyses performed in the WT and KO mice revealed that SFTSV induced red blood cell leakage and hyperemia in the liver, pulmonary, spleen and kidney, which were alleviated in the respective organs of the KO mice (Fig. 7G). We also determined the expression of sVEGFR1 in the spleens by qPCR and showed that the expression levels of sVEGFR1 in the KO mice decreased significantly as compared with the WT mice (Fig. 7H). At the same time, the protein level of sVEGFR1 and SFTSV-NP in spleens were both decreased in the KO mice (Fig. 7I). Consistently, immuno-fluorescence analysis showed that sVEGFR1 and SFTSV-Gn decreased in the spleens of the KO mice (Fig. 7J). Collectively, these results demonstrated that sVEGFR1 expression were reduced in EGR1 KO mice during SFTSV infection, alleviating angiogenesis dysfunction. Notably, EGR1 KO mice also exhibited lower levels of SFTSV NP protein and viral RNA, suggesting that EGR1 KO mice were less susceptible to SFTSV.

## Discussion

Hemorrhage is the hallmark of SFTS and 35% patients with a SFTSV infection had hemorrhagic symptom, which is higher than that reported for other viral hemorrhagic fevers, such as Ebola virus disease or dengue (Li et al, 2018a; Simmons et al, 2012; Team et al, 2015) and considered as a critical pathophysiological manifestation in severe infections, which often resulted in death (Gai et al, 2012; Li et al, 2021b; Saijo, 2018). Current opinions consider thrombo-cytopenia as the key pathological manifestation of SFTSV infection (Fang et al, 2023); however, the impairment of blood clotting resulted from thrombocytopenia is not sufficient to lead to hemorrhage without the disruption of vascular endothelial integrity. Little information is available on how SFTSV infection causes disruption of vascular endothelium. We speculate that virus-

mediated disruption of vascular endothelial integrity likely plays a role and it is thus important to investigate the hemorrhagic mechanism of SFTSV infection. Tissue and vascular repair are dependent on the formation of new microvessels. The process of angiogenesis is regulated by a balance between pro-angiogenic and anti-angiogenic factors, and the shift to an anti-angiogenic phenotype is a key event in hemorrhagic fever infectious diseases. Restoring this balance is a potential way to improve hemorrhagic manifestations. In the present study, we showed that the disruption of this balance was responsible for the deleterious consequence of the endovascular vessels during SFTSV infection, which likely resulted in the hemorrhage.

In the current study, we showed that SFTSV infection upregulated sVEGFR1 expression and secretion in monocytes/macrophages (Figs. 1 and EV2), consistent with previous reports that sVEGFR1 protein is highly expressed in macrophages (Shibuya, 2006; Tsao et al, 2007; Wu et al, 2015; Zhao et al, 2017). More significantly, our findings demonstrated that SFTSV infection induced de novo synthesis and subsequent release of sVEGFR1 in vivo (Fig. 2). We further showed that sVEGFR1 was elevated in SFTS patients, which was significantly correlated with disease severity (Fig. 2E). Furuta et al also observed a significant increase in sVEGFR1 levels in dengue shock syndrome patients, which suggests that activation of monocytes/macrophages by the dengue virus leads to increased expression of soluble and surface VEGFR1 on cells during severe dengue infection (Furuta et al, 2012).

A common feature of all viral hemorrhagic fever is the systemic dissemination of the virus into the monocytes/macrophages. Previous findings suggested that the macrophage derived-sVEGFR1 was a regulator of vascular response of endothelial cells, contributing to impaired angiogenesis in aged mice (Zhao et al, 2017). The dominant factor in angiogenesis is VEGFA which binds the pro-angiogenic VEGF-receptor 1 (VEGFR1), a kinase-impaired receptor with higher affinity and lower tyrosine kinase activity for VEGFA than VEGF-receptor 2 (VEGFR2) (Shibuya, 2013). VEGFR1 also exists in a soluble form lacking the transmembrane and cytoplasmic domains. Release of sVEGFR1 provides a negative feedback loop regulating angiogenesis by sequestering VEGFA and thereby preventing it from binding to VEGFRs. For instance, overexpressed sVEGFR1 has a prognostic value in hemorrhagic

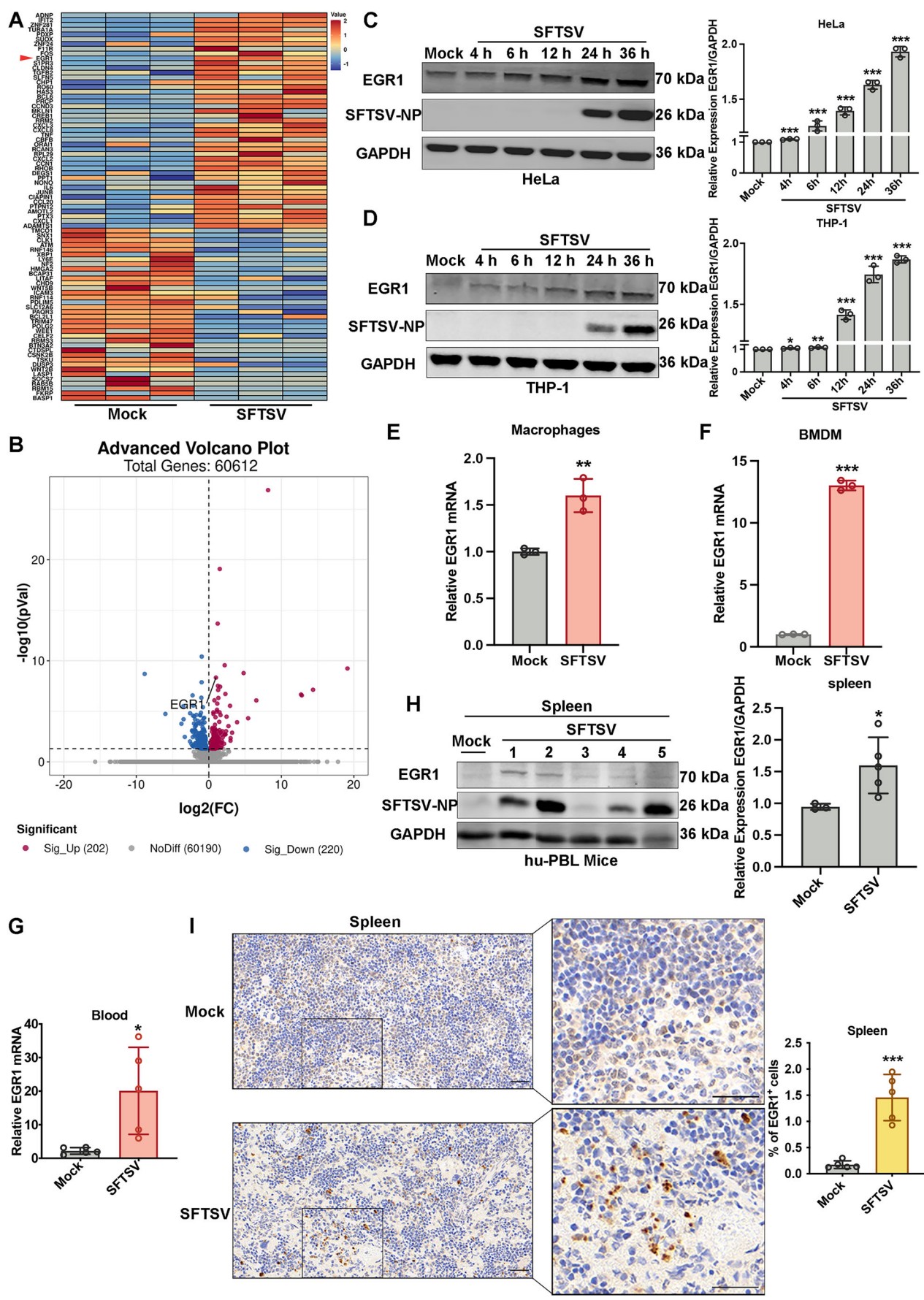

**Figure 4.  SFTSV infection up-regulates EGR1.**

(A) RNA-seq was performed in SFTSV-infected or mock-infected HUVECs. The heatmap showed that the 81 genes relating to angiogenesis were among 422 genes with significant differences (based on an adjusted $P$ value ≤ 0.05) in expression after SFTSV infection. (B) Comparison of gene expression of SFTSV-infected or mock-infected HUVECs. Red dots represent genes with higher expression in SFTSV-infected cells and blue dots represent genes with higher expression in mock-infected cells (based on an adjusted $P$ value ≤ 0.05). (C) HeLa cells were infected with SFTSV for 4, 6, 12, 24 and 36 h. Intracellular EGR1 protein was analyzed by western blotting (left). The EGR1 expression relative to GAPDH was shown (right). $n = 3$ biological replicates. Statistical significance was determined by two-tailed unpaired Student $t$ test. (***$P = 0.0010$, ***$P = 0.0006$, ***$P < 0.0001$, ***$P < 0.0001$, ***$P < 0.0001$). (D) are the same experiments as in (C), but evaluated in THP-1 cells. $n = 3$ biological replicates. Statistical significance was determined by two-tailed unpaired Student $t$ test. (*$P = 0.0281$, **$P = 0.0053$, ***$P < 0.0001$, ***$P < 0.0001$, ***$P < 0.0001$). (E) Primary human monocyte-derived-macrophages were infected with SFTSV (MOI = 1) for 24 h. EGR1 mRNA was measured by qPCR. $n = 3$ biological replicates. Statistical significance was determined by two-tailed unpaired Student $t$ test. (**$P = 0.0046$). (F) BMDMs from C57BL/6J mice were matured to macrophage and then infected with SFTSV (MOI = 1) for 24 h. BMDMs were infected with SFTSV for 24 h. EGR1 mRNA was measured by qPCR. $n = 3$ biological replicates. Statistical significance was determined by two-tailed unpaired Student $t$ test. (***$P < 0.0001$). (G) hu-PBL NCG mice were infected with SFTSV ($n = 5$) at $1 \times 10^3$ TCID$_{50}$ or mock-infected ($n = 5$), and blood samples were collected. EGR1 mRNA in blood cells was determined by qPCR. $n = 5$ biological replicates. Statistical significance was determined by two-tailed unpaired Student $t$ test. (*$P = 0.0152$). (H) Infected mice ($n = 5$) were euthanized and spleens were collected. The indicated proteins in spleens were measured by western blotting (left). Numeric label indicates individual mice. The EGR1 protein expression relative to GAPDH was shown (right). $n = 3$ biological replicates for Mock; $n = 5$ biological replicates for SFTSV. Statistical significance was determined by two-tailed unpaired Student $t$ test. (*$P = 0.049$). (I) Detection of EGR1 by immunohistochemistry in the spleen sections after SFTSV infection (left). Shown on the right are subset images of the regions outlined by rectangles on the left. Scale bar = 50 μm. The quantifications of relative EGR1 intensities were determined by ImageJ software (right). $n = 5$ biological replicates. Statistical significance was determined by two-tailed unpaired Student $t$ test. (***$P = 0.0002$). Data information: Data shown are mean ± SD of at least three biological replicates with each data point representing a biological experiment. (*$P < 0.05$; **$P < 0.01$; ***$P < 0.001$). Source data are available online for this figure.

fever infectious diseases (Bakir et al, 2013; Seet et al, 2009). Growing evidences have shown the important role of sVEGFR1 in regulating angiogenesis in viral hemorrhagic diseases and the function of sVEGFR1 as a biomarker in different diseases. A previous study had shown that sVEGFR1 was increased in Crimean-Congo Hemorrhagic Fever (CCHF) patients, and it might be an important biomarker to determine the risk of disease severity and death in CCHF patients (Bakir et al, 2013). Additional study had demonstrated that sVEGFR1 levels were significantly increased in patients with dengue hemorrhagic fever compared with the levels in those with dengue fever (Seet et al, 2009). However, these earlier studies did not address the functional roles of elevated sVEGFR1 with respect to the pathophysiology of the infections. Our study not only assessed endothelial dysfunction caused by elevated serum sVEGFR1 and of its potential impact on vascular endothelial integrity but also demonstrated that sVEGFR1 was a potential therapeutic target as removing sVEGFR1 markedly mitigated the disruption of vascular endothelial integrity, as shown by tube formation assays, aortic ring assays and Matrigel plug assays (Fig. 3; Appendix Fig. S1). The efficiency of sVEGFR1 removal was verified by ELISA and the results confirmed a substantial reduction in sVEGFR1 levels in the sera after treatment (Appendix Fig. S2). In addition, there was also statistically significant difference for sVEGFR1 between healthy individuals and hemorrhagic fever with renal syndrome (HFRS) patients (Appendix Fig. S3A). HFRS is a viral hemorrhagic fever caused by infection by Hantaan virus (HTNV), which also belongs to the *Bunyaviridae* family. We also showed the role of HTNV-induced sVEGFR1 in angiogenesis by performing a tube formation assay on HUVECs using sera collected from HFRS patients and the tubule formation was also significantly decreased in these groups compared with the healthy group (Appendix Fig. S3B).

The intricate balance between sVEGFR1 and VEGFA maintains a subtle homeostasis of angiogenesis to check the endovascular repair/growth and prevention of overgrowth. The regulatory element of sVEGFR1 is EGR1, a nucleic residing early growth response gene-1 (EGR1) that can act directly on VEGFR1 regulatory element (Vidal et al, 2000). EGR1 is a transcription factor that is mainly involved in the processes of tissue injury,

immune responses, viral infection and tumor anti-angiogenic action. EGR1 also plays roles in viral pathogenesis (Dahal et al, 2020; Kim et al, 2007; Romagnoli et al, 2008; Sarkar and Verma, 2017; Wong et al, 2022; Yao et al, 2012).During viral infections, EGR1 expression can facilitate either an antiviral or pro-viral cellular state, depending on the virus and the transcriptional events regulated by EGR1. Infection of astrocytes with Rift Valley fever virus (RVFV), a hemorrhagic fever virus known to induce significant hepatitis and encephalitis (Gaudreault et al, 2019), induced a robust induction of EGR1 mRNA (Lehman et al, 2022). EGR1 might also play a positive role in viral replication. SFTSV infection promoted the EGR1 expression by as much as 2-fold in vitro and in vivo (Fig. 4). Our observations revealed similar results to other studies, underscoring the biological relevance and general importance of EGR1 in viral infections.

Previous studies demonstrated that VEGFR1 promoter sequence presented a binding site for EGR1 at positions −24 to −16, and EGR1 was able to bind to this DNA sequence, resulting in enhanced transcriptional activity (Vidal et al, 2000). This is consistent with our findings that EGR1 had a positive role in sVEGFR1 production which inhibited angiogenesis during SFTSV infection (Fig. 5). Nonetheless, the specific mechanism whereby EGR1 modulates these processes and the potential viral factor modulating EGR1 remain to be elucidated. In addition, we found that sVEGFR1 expression was significantly lower in EGR1 KO mice during SFTSV infection compared to WT mice, and viral load was also significantly lower in EGR1 KO mice (Fig. 7), suggesting that EGR1 is important for SFTSV infection, which is consistent with our previous findings (Jiang et al, 2024).

The effective therapeutic window is rather narrow for SFTSV infection as many patients are misdiagnosed initially due to the early SFTS symptoms' similarity to many other viral infections, such as Influenza. Once diagnosed, patients, particularly those elderly and with underlining medical conditions, tend to present poor prognoses. We and others have shown that old age was a high-risk factor for SFTS and many deaths occurred in the elderly (Song et al, 2018). We have reported a therapeutic humanized nanobody (VHH) that potently inhibited SFTSV in vitro and

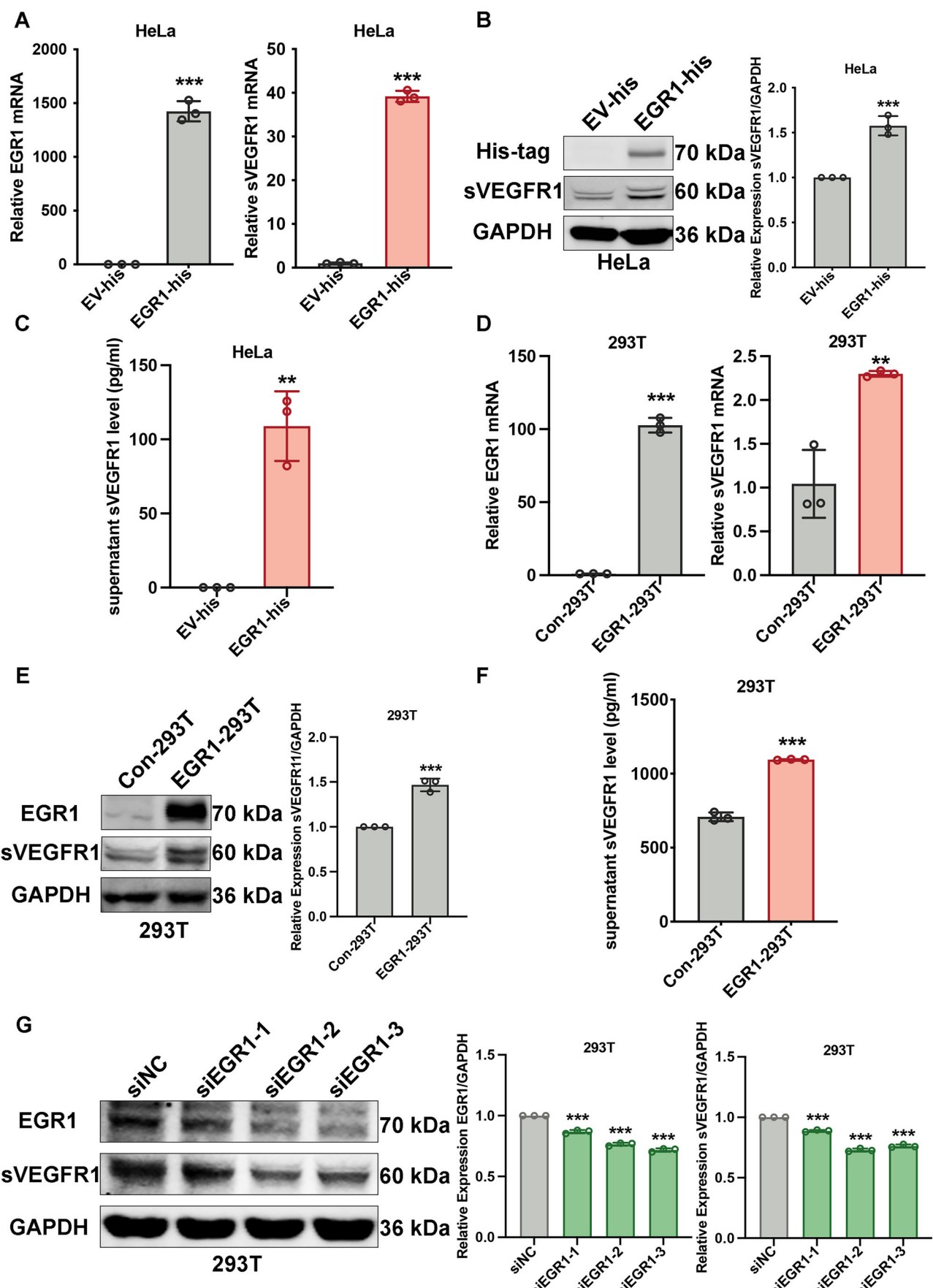

**Figure 5. EGR1 positively regulates the expression of sVEGFR1.**

(A) HeLa cells were transiently transfected with a plasmid expressing EGR1 or an empty vector (EV-his). After 36 h post transfection, the relative EGR1 mRNA (left) and sVEGFR1 mRNA (right) in HeLa cells were measured by qPCR. $n = 3$ biological replicates. Statistical significance was determined by two-tailed unpaired Student $t$ test. (EGR1: ***$P < 0.0001$; sVEGFR1: ***$P < 0.0001$). (B) The expression of sVEGFR1 was measured by western blotting (left) and quantified by density of the WB bands (right). $n = 3$ biological replicates. Statistical significance was determined by two-tailed unpaired Student $t$ test. (***$P = 0.0008$). (C) sVEGFR1 protein level in HeLa cell culture supernatant was measured by ELISA. $n = 3$ biological replicates. Statistical significance was determined by two-tailed unpaired Student $t$ test. (**$P = 0.0013$). (D) HEK-293T cells were infected with a recombinant lentivirus vector (pLV-Puro-EGR1) and screened with puromycin for cell clones with stable EGR1 overexpression (EGR1-293T cells). EGR1-293T cells, along with HEK-293T cells infected with pLV-Puro vector (Con-293T cells), were tested for the mRNA levels of EGR1 and sVEGFR1 by qPCR. $n = 3$ biological replicates. Statistical significance was determined by two-tailed unpaired Student $t$ test. (EGR1: ***$P < 0.0001$; sVEGFR1: **$P = 0.0050$). (E) The protein levels of EGR1 and sVEGFR1 in EGR1-293T cells and Con-293T cells were measured by western blotting (left) and quantified by density of the WB bands (right). $n = 3$ biological replicates. Statistical significance was determined by two-tailed unpaired Student $t$ test. (***$P = 0.0003$). (F) sVEGFR1 protein in cell culture supernatant was measured by ELISA. $n = 3$ biological replicates. Statistical significance was determined by two-tailed unpaired Student $t$ test. (***$P < 0.0001$). (G) Cells transfected with siNC or siEGR1 (siEGR1-1, siEGR1-2 and siEGR1-3) were harvested at 36 hpt to evaluate EGR1 protein knock-down efficiency and sVEGFR1 protein levels by western blotting. $n = 3$ biological replicates. Statistical significance was determined by two-tailed unpaired Student $t$ test. (EGR1: ***$P < 0.0001$, ***$P < 0.0001$, ***$P < 0.0001$; sVEGFR1: ***$P < 0.0001$, ***$P < 0.0001$, ***$P < 0.0001$). Data information: Data shown are mean ± SD of three biological replicates. (**$P < 0.01$; ***$P < 0.001$). Source data are available online for this figure.

in vivo (Wu et al, 2020), which are currently under development as a therapeutic drug. However, as with many acute viral infections, targeting replicating virus has to be early, as shown by our in vivo evaluation of the nanobody in a hu-PBL NCG mouse model (Wu et al, 2020). Hemorrhage, as a later pathogenic event and often associated with eventual death (Feldmann and Geisbert, 2011; Messaoudi and Basler, 2015; Zhang et al, 2012), may serve as a potential late-stage intervention for SFTS. We conducted additionally experiment where exogenous sVEGFR1 was added directly to the Matrigel in the absence of human serum and the results showed no significant inhibition of tube formation, indicating that additional factors present in the human serum are likely required to mediate the anti-angiogenic effect observed in our study (Appendix Fig. S4). We showed that SFTSV infection upregulated VEGFA expression and secretion (Appendix Fig. S5A,B), but did not affect sVEGFR1 secretion in HUVECs (Appendix Fig. S5C). sVEGFR1, induced by SFTSV and antagonized VEGFA, is a readily accessible target and by utilizing neutralizing antibodies the vascular endothelium was repaired by VEGFA, as shown in the current study. sVEGFR1 neutralizing antibodies may serve as a possible adjunctive therapy in the SFTSV infection. We found that sVEGFR1 neutralizing antibody restored angiogenesis in vivo during SFTSV infection (Fig. 6). We previously reported that SFTSV downregulated vascular endothelial cadherin (VE-cadherin), a critical component of adherent junction, by inducing phosphorylation of VE-cadherin at Tyr658 and internalization of the p-VE-cadherin (Xu et al, 2021), thus disrupting adherent junctions and endovascular integrity. Li XK et al also observed a significant decrease in VE-cadherin levels in SFTSV infection (Li et al, 2018b). We observed in vitro that exogenous sVEGFR1 added to HUVEC downregulated the cellular VE-cadherin, which was reversed by the treatment of a sVEGFR1 neutralizing antibody, as shown in Appendix Fig. S6.

In conclusion, we reveal a distinct molecular mechanism by which SFTSV-induced sVEGFR1 release via upregulated EGR1 contributes to hemorrhage, suggesting that SFTSV infection disrupts the balance between the pro-angiogenic and anti-angiogenic state. The excessive secretion of sVEGFR1 then antagonizes VEGFA, thus preventing the repair of the disrupted vascular endothelium caused by SFTSV infection. Thus, the EGR1/sVEGFR1/hemorrhage axis can serve as a potential target for the development of novel anti-hemorrhagic therapies for SFTSV infection (Fig. 8).

## Methods

### Reagents and tools table

| Reagent/resource | Reference or source | Identifier or catalog number |
| --- | --- | --- |
| **Experimental models** | | |
| HEK-293T cells | ATCC | Cat#: BFN60700191 |
| HUVEC cells | ATCC | Cat#: BNF607200285 |
| HeLa cells | ATCC | Cat#: BFN60700111 |
| THP-1 cells | ATCC | Cat#: BNF60700157 |
| NCG mice | GemPharmatech | Cat#: T001475 |
| EGR1 KO mice | GemPharmatech | Cat#: T0014356 |
| **Recombinant DNA** | | |
| pCMV3-C-His | provided by our laboratory | |
| pCMV3-EGR1-C-His | SinoBiological | Cat#: HG11870-AH |
| pLV-puro-EGR1-C-GFP | SinoBiological | Cat#: HG11870-ACGLN |
| pLV-puro-C-GFP | provided by our laboratory | |
| pLV-puro | provided by our laboratory | |
| psPAX2 | provided by our laboratory | |
| pMD2.G | provided by our laboratory | |
| **Antibodies** | | |
| Mouse anti-SFTSV NP | Cambridge Bio | Cat#: 1-05-0130 |
| Rabbit anti-sVEGFR1 | Proteintech | Cat#: 13687-1-AP |
| Mouse-anti-GAPDH | Proteintech | Cat#: 60004-1-Ig |
| Rabbit anti-EGR1 | Proteintech | Cat#: 22008-1-AP |
| Rabbit anti-EGR1 | Abclonal | Cat#: A7266 |
| anti-His tag | Proteintech | Cat#: 66005-1-Ig |
| Rabbit anti-SFTSV Gn | provided by our laboratory | |
| Camel anti-SFTSV Gn | provided by our laboratory | |
| Mouse anti-sVEGFR1 | R&D | Cat#: AF471 |

| Reagent/resource | Reference or source | Identifier or catalog number |
|---|---|---|
| Goat IgG Control | R&D | Cat#: AB-108-C |
| Human anti-sVEGFR1 | R&D | Cat#: AF321 |
| Mouse IFNAR1 blocking antibody | Selleck | Cat#: A2121 |
| Rabbit anti-CD31 | Proteintech | Cat#: 11265-1-AP |
| Goat anti-M800 | LI-COR | Cat#: 926-32210 |
| Goat anti-M680 | LI-COR | Cat#: 926-68070 |
| Goat anti-R800 | LI-COR | Cat#: 926-32211 |
| Goat anti-R680 | LI-COR | Cat#: 926-68071 |
| **Oligonucleotides and other sequence-based reagents** | | |
| qPCR primers for SFTSV L segment-F | This study | 5'-TGGTGGATGTCATAGAAGGC-3' |
| qPCR primers for SFTSV L segment-R | This study | 5'-GAGTAATCCTCTTGCCTGCT-3' |
| qPCR primers for human sVEGFR1-F | This study | 5'-GGCTGTTTTCTCTCGGATCTC-3' |
| qPCR primers for human sVEGFR1-R | This study | 5'-CATCTCCTCCGAGCCTGAAAG-3' |
| qPCR primers for mouse sVEGFR1-F | This study | 5'-ACACCGCGGTCTTGCCTTAC-3' |
| qPCR primers for mouse sVEGFR1-R | This study | 5'-CAGCCTTTTGTCCTCCTGGC-3' |
| qPCR primers for human EGR1-F | This study | 5'-AGCAGCACCTTCAACCCTCAGG-3' |
| qPCR primers for human EGR1-R | This study | 5'-GAGTGGTTTGGCTGGGGTAACT-3' |
| qPCR primers for mouse EGR1-F | This study | 5'-AGCGAACAACCCTATGAGCACC-3' |
| qPCR primers for mouse EGR1-R | This study | 5'-ATGGGAGGCAACCGAGTCGTTT-3' |
| qPCR primers for human GAPDH-F | This study | 5'-GAGTCAACGGATTTGGTCGT-3' |
| qPCR primers for human GAPDH-R | This study | 5'-CTTGATTTTGGAGGGATCTCGC-3' |
| qPCR primers for mouseβ-actin-F | This study | 5'-AACAGTCCGCCTAGAAGCAC-3' |
| qPCR primers for mouseβ-actin-R | This study | 5'-CGTTGACATCCGTAAAGACC-3' |
| PCR primers for Egr1-ko-F (primer 1-F) | This study | 5'-CCTGGATTGGAGGGCAACTGA-3' |
| PCR primers for Egr1-ko-R (primer 1-R) | This study | 5'-GTTCTCACCTGTTCCAAGGCC-3' |
| PCR primers for Egr1-wt-F (primer 2-F) | This study | 5'-CCACCATAGGCCTTACTCAGCA-3' |
| PCR primers for Egr1-wt-R (primer 2-R) | This study | 5'-GACTTTTGGATGGGCACATGAC-3' |
| siEGR1-1-as | This study | UUUCUUGUCCUUCUGCCGCUU |
| siEGR1-1-ss | This study | GCGGCAGAAGGACAAGAAAUU |
| siEGR1-2-as | This study | UUGGUCAUGCUCACUAGGCUU |
| siEGR1-2-ss | This study | GCCUAGUGAGCAUGACCAAUU |
| siEGR1-3-as | This study | UUAGGGUAGUUGUCCAUGGUU |
| siEGR1-3-ss | This study | CCAUGGACAACUACCCUAAUU |

| Reagent/resource | Reference or source | Identifier or catalog number |
|---|---|---|
| **Chemicals, enzymes and other reagents** | | |
| Recombinant human sVEGFR1 | R&D | Cat#: 321-FL |
| Recombinant human VEGFA | Proteintech | Cat#: HZ-1038 |
| Human sVEGFR1 ELISA Kit | Proteintech | Cat#: KE00113 |
| Mouse sVEGFR1 ELISA Kit | BOSTER | Cat#: EK0589 |
| Matrigel | ABW Bio | Cat#: 0827265 |
| Matrigel | ABW Bio | Cat#: 082706 |
| Matrigel | ABW Bio | Cat#: 082703 |
| Lipo8000 | beyotime | Cat#: C0533 |
| Lipofectamine 3000 | ThermoFisher Scientific | Cat#: L3000015 |
| **Software** | | |
| GraphPad Prism 9.0 | San Diego, CA, USA | https://www.graphpad.com/scientific-software/prism/ |
| **Other** | | |

## Cells and viruses

HEK-293T, HUVEC, HeLa and Vero E6 cells were cultured and maintained in Dulbecco's modified Eagle's medium (DMEM, Gibco, USA) supplemented with 10% fetal bovine serum (ExCell Bio, Shanghai, China) in a humidified 37 °C 5% $CO_2$ incubator. HUVECs were obtained from the American Type Culture Collection (ATCC, USA). HUVECs (Passages 3–6) were acclimatized for 1–2 days either in working media (DMEM with 10% FBS) or starving media (DMEM without FBS) before the experimental start. THP-1 cells were maintained in RPMI-1640 (Gibco, USA) medium supplemented with 10% FBS in a humidified 37 °C 5% $CO_2$ incubator. SFTSV of subtype E-JS-2013-24 was provided by Jiangsu Provincial CDC, Nanjing, China. The virus was passaged in Vero E6 cells (Song et al, 2018). $TCID_{50}$ was determined using methods described previously (Xu et al, 2021). Camel and Rabbit antibodies against SFTSV-Gn were obtained by immunization with soluble SFTSV Gn protein expressed in mammalian cell line and donated by Y-Clone, Ltd.

## Monocytes-derived macrophages differentiation

Human peripheral blood mononuclear cells (PBMCs) (derived from healthy donors visiting the Drum Tower Hospital, Nanjing University) were isolated by centrifugation over a Ficoll-Paque (GE Healthcare) density gradient. CD14+ monocytes were further purified by positive selection using CD14 microbeads (Miltenyi). Monocytes were differentiated over 7 days into macrophages in RPMI-1640 with 10% heat-inactivated FBS in the presence of 50 ng/ml recombinant human M-CSF. Macrophages were washed twice with serum-free RPMI and then infected with SFTSV in RPMI-1640 with 10% heat-inactivated FBS.

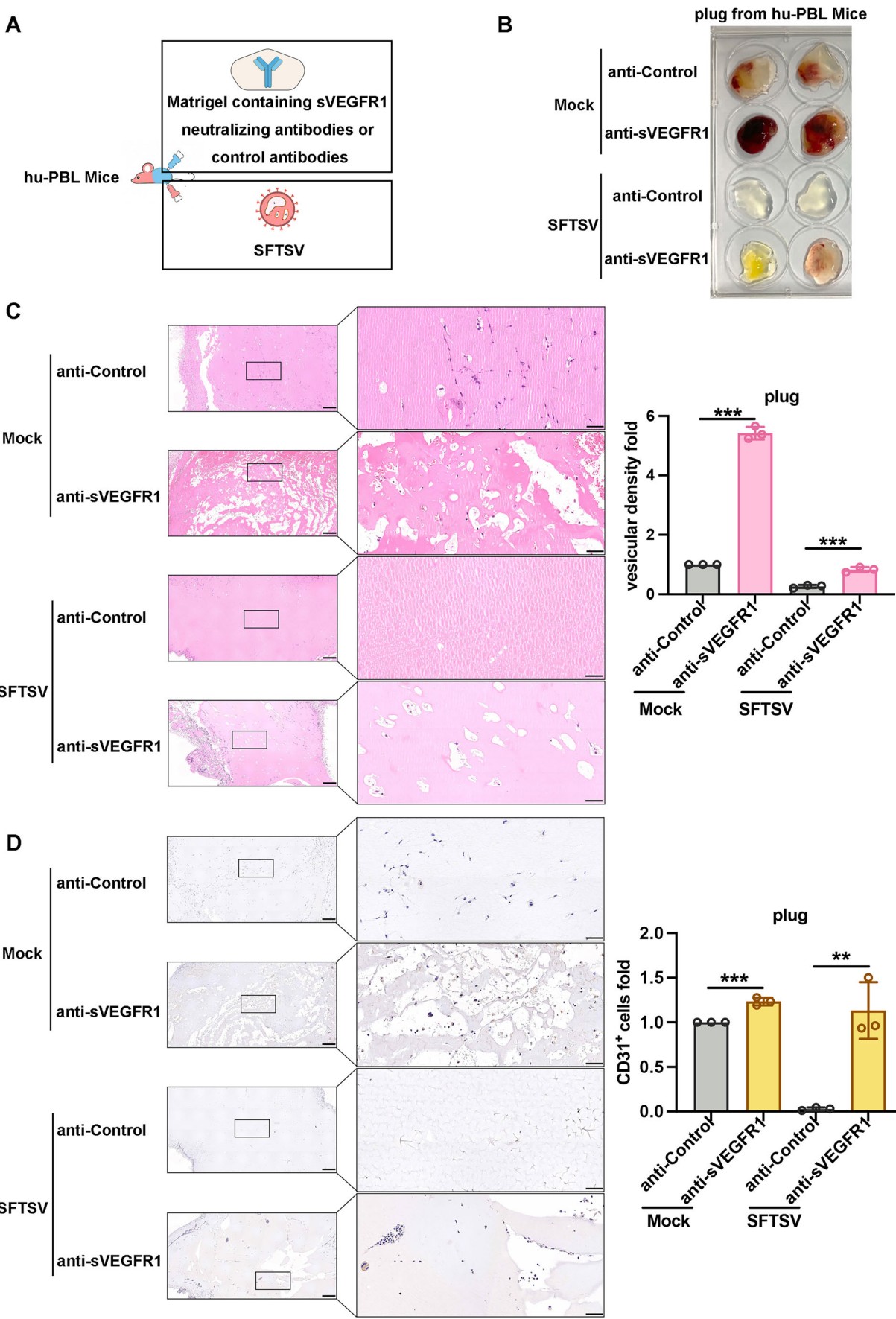

**Figure 6. sVEGFR1 neutralizing antibodies restore angiogenesis in a Matrigel plug embedded in SFTSV-infected mice in vivo.**

(A) Schematic representation of experimental design. The Matrigel plug assays were performed by subcutaneous injection of Matrigel containing sVEGFR1 Ab or Control Ab into hu-PBL NCG mice followed with or without SFTSV infection. Seven days after infection, the Matrigel plugs were removed and photographed. (B) Images of Matrigel plugs containing control Ab (10 μg/ml) or sVEGFR1 Ab (10 μg/ml) recovered from mice dorsum after the mice were infected or uninfected with SFTSV. (C) Representative cross-sections of H&E-staining of the Matrigel plugs, harvested as described in (B). Shown on the right are subset images of the regions outlined by rectangles on the left. Scale bars, 250 μm (left) and 50 μm (right), respectively. $n = 3$ biological replicates for each group. Statistical significance was determined by two-tailed unpaired Student $t$ test. (Mock, ***$P < 0.0001$; SFTSV, ***$P = 0.0006$). (D) After the plugs were embedded in wax and sectioned, the sections were immune-stained with CD31 antibodies. Representative images of immunostaining are shown. Shown on the right are subset images of the regions outlined by rectangles on the left. Scale bars, 250 μm (left) and 50 μm (right), respectively. $n = 3$ biological replicates for each group. Statistical significance was determined by two-tailed unpaired Student $t$ test. (Mock, ***$P = 0.0007$; SFTSV, **$P = 0.0039$). Data information: Data shown are mean ± SD of three biological replicates. (**$P < 0.01$; ***$P < 0.001$). Source data are available online for this figure.

## Isolation and culture of murine bone marrow-derived macrophages (BMDMs)

Bone marrow-derived macrophages (BMDMs) from 7- to 9-week-old C57BL/6J mice were obtained by in vitro differentiation of the primary femur and tibia BM cells. Briefly, femurs and tibiae from wild-type mice were dissected, cleaned, disinfected in 70% ethanol, and washed with fully supplemented RPMI-1640 medium. After lysing erythrocytes, the BM cells were then cultured in a Petri dish (at a density of $4 \times 10^6$ cells/10-cm dish) supplemented with 30% L929 cell-conditioned medium. Cells were matured to macrophage over 5–6 days. Adherent cells were recovered and infected with SFTSV for the experiments.

## SFTSV infection of hu-PBL NCG mice

Immunodeficient NCG mice was purchased from GemPharmatech (Nanjing, China). Similar to NOD.Cg-Prkd$^{cscid}$ Il2rg$^{tm1Wjl}$/SZJ mice, the NCG mice lacked the IL-2 receptor gene in a SCID background, resulting in the absence of murine T and B cells and very small numbers of NK cells. Hu-PBL mice were generated as described previously (Nakata et al, 2005). Briefly, $2 \times 10^7$ human PBMCs were injected intraperitoneally (i.p.) into each of 4- to 6-week-old NCG mice. PBMC engraftments were evaluated at the 21st day after the transplantation by FACS. Only mice with comparable engraftment levels (Appendix Fig. S7) were included and randomly divided into uninfected control or infected groups. Hu-PBL mice ($n = 5$) were injected intraperitoneally with SFTSV ($1 \times 10^3$ TCID$_{50}$) and mock-infected mice ($n = 5$) were injected intraperitoneally with the same volume of DMEM as controls. Spleen and blood cells were collected at 14 days post infection at the time of sacrifice (Xu et al, 2021).

## Quantitative real-time PCR (qPCR)

Total RNA was extracted using TRIzol reagent (Life Technologies, USA) and reverse transcribed using PrimeScript RT master mix for real-time-PCR (TaKaRa). Quantitative real-time PCR (qPCR) was performed using ABI SYBR green master mix (Life Technologies) on an ABI Prism 7500 sequence detection system (Applied Biosystems). The glyceraldehyde-3-phosphate dehydrogenase (GAPDH) and beta-actin (β-actin) gene were used for normalization of mRNA including SFTSV viral RNA, EGR1 and sVEGFR1 quantification. All reactions were carried out in triplicate, and relative transcript levels were calculated using the $2^{\Delta\Delta Ct}$ method. All the sequences of primers used in this study are listed in the Reagents and Tools Table.

## Western blotting analysis

For Western blotting, the cells were collected at the indicated time points. Whole-cell lysates were prepared in RIPA lysis buffer and measured for protein concentration by BCA protein assay kit (Life Technologies, USA) to equalize protein loading. After electrophoresis, proteins were transferred onto a polyvinylidene difluoride (PVDF) membrane (Millipore, USA). The membranes were then blocked for 1 h at room temperature in 2% bovine serum albumin (BSA) and then incubated with primary antibodies at an appropriate dilution overnight at 4 °C. The membranes were then incubated with secondary antibodies. Protein bands were visualized using an Odyssey Imaging System (Li-COR Bioscience). Primary antibodies included antibodies specific for sVEGFR1, EGR1, and GAPDH (Proteintech, Wuhan, China), antibodies specific for SFTSV-NP (Cambridge Biologics, Suzhou, China) and Rabbit anti-SFTSV-Gn antibodies were kindly provided by Y-Clone, Ltd. Secondary antibodies includes IRDye Fluor 800-labeled IgG or IRDye Fluor 680-labeled IgG secondary antibody (Li-COR Bioscience).

## Enzyme-linked immunosorbent assay (ELISA)

Cells were seeded into six-well plate with a density of $2.0 \times 10^4$ per well. Cell supernatant was obtained after centrifugation. The concentration of sVEGFR1 in cell supernatant or human serum was determined with human sVEGFR1 ELISA kit (Proteintech, Wuhan, China) according to the manufacturer's instruction. The concentration of sVEGFR1 in mice serum was determined with mouse sVEGFR1 ELISA kit (Boster Biotechnology, Wuhan, China) according to the manufacturer's instruction. All the samples were analyzed in triplicate. The concentrations of sVEGFR1 in the samples were determined from standard curve.

## Subjects

The SFTS patients who were treated in Nanjing Drum Tower Hospital during 2021-2022 were included in this study. SFTSV infection was confirmed by qPCR tests or serological tests. Severe SFTS patients were defined by the presence of hemorrhagic manifestations (melena, hematemesis, hemoptysis, ophthalmorrhagia and gingival bleeding), any one or more organ failure or encephalitis development (Lu et al, 2015). Healthy blood donors with comparable age and gender who were determined to be SFTSV negative by both qPCR tests and serological tests were enrolled as controls from the department of physical examination in the same hospital. Serum specimens were stored at −80 °C until used.

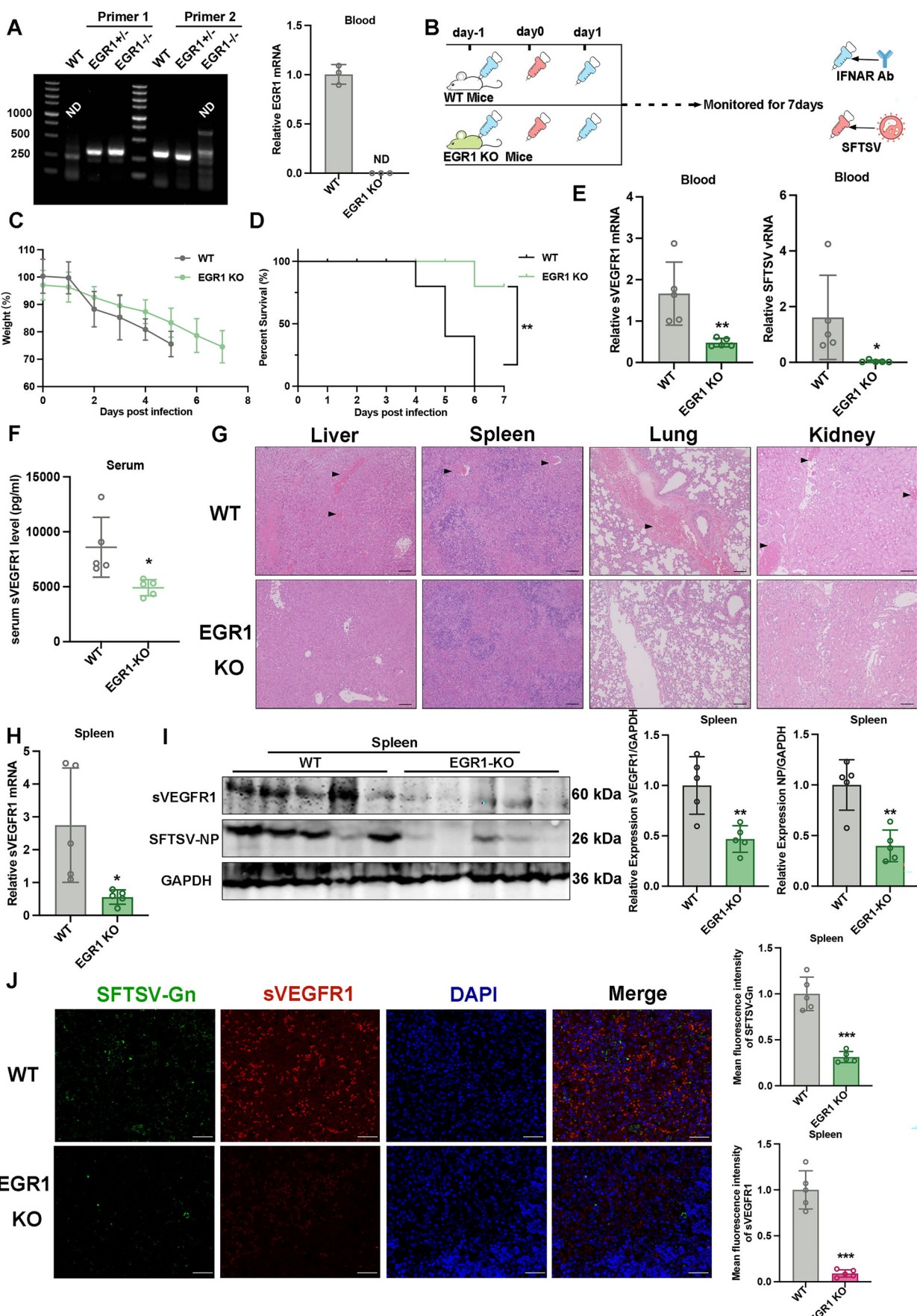

Figure 7.  sVEGFR1 expression is reduced in EGR1 KO mice during SFTSV infection, alleviating angiogenesis dysfunction.

(A) PCR (left) and q-PCR (right) were used to detect the genotype and KO efficiency, respectively ($n = 3$ biologically independent WT mice and KO mice). ND, Not Detected. (B) Experimental scheme of the SFTSV-infected mouse model. WT and EGR1-KO mice were infected with SFTSV at $1 \times 10^5$ TCID$_{50}$, respectively. Liver, spleen, lung and kidney were isolated from two groups for analysis. (C, D) The percent of body weight and survival changes were monitored for 7 days. $n = 5$ biological replicates for each group. Statistical significance was determined by Log-rank test. ((D): **, $P = 0.0048$) (E) sVEGFR1 (left) and SFTSV viral RNA mRNA (right) in blood cells from the WT and KO mice ($n = 5$) were determined by qPCR. $n = 5$ biological replicates. Statistical significance was determined by two-tailed unpaired Student $t$ test. (sVEGFR1: **$P = 0.0087$; SFTSV: *$P = 0.0481$) (F) The serum level of sVEGFR1 protein was measured by capture ELISA. Each point represents an individual. $n = 5$ biological replicates. Statistical significance was determined by two-tailed unpaired Student $t$ test. (*$P = 0.0193$). (G) Histopathological examinations of the tissues collected from the WT and KO mice. Tissues were collected from mice on day 4. The representative photographs of H&E were shown. The black arrows indicate hyperemia. Bars, 100 µm. (H) The infected mice were euthanized, and spleens were collected. sVEGFR1 mRNA in the spleens from the WT and KO mice were determined by qPCR. $n = 5$ biological replicates. Statistical significance was determined by two-tailed unpaired Student $t$ test. (*$P = 0.0235$). (I) sVEGFR1 and the viral NP protein in spleens were measured by western blotting (left). The sVEGFR1 and SFTSV-NP protein expression relative to GAPDH was shown (right), respectively. $n = 3$ biological replicates for Mock; $n = 5$ biological replicates for SFTSV. Statistical significance was determined by two-tailed unpaired Student $t$ test. (sVEGFR1: **$P = 0.0054$; NP: **$P = 0.0018$). (J) Detection of sVEGFR1 and viral Gn protein by immunofluorescence in the spleens after SFTSV infection (left). SFTSV-Gn (green), sVEGFR1 (red) and nuclei (DAPI, blue) were visualized using either antibodies specific for Gn or sVEGFR1, or DAPI for nuclei. A merged image of green, red and blue channels is shown (merge). The quantifications of relative SFTSV-Gn and sVEGFR1 intensities were determined by ImageJ software (right). Scale bars, 25 µm. $n = 5$ biological replicates. Statistical significance was determined by two-tailed unpaired Student $t$ test. (Gn: ***$P < 0.0001$; sVEGFR1: ***$P < 0.0001$). Data information: Data shown are mean ± SD of at least three biological replicates with each data point representing a biological experiment. (*$P < 0.05$; **$P < 0.01$; ***$P < 0.001$). Source data are available online for this figure.

Informed consent was obtained from all subjects, in accordance with the Declaration of Helsinki and principles set out in the Department of Health and Human Services Belmont Report, and the research was approved by the Ethics Committee of Nanjing Drum Tower Hospital.

## RNA sequencing analysis

To identify the expressed differentiation of genes between SFTSV-infected group and mock-infected group, we performed RNA sequencing (RNA-Seq) analysis at LC-BIOTECHNOLOGIS (HANGZHOU, CO., LTD). Total RNA was extracted using TRIzol reagent following the manufacturer's procedure. Total RNAs from three biological replicates were pooled for cDNA library construction. cDNA library preparation and sequencing were carried out on an Illumina NovaSeq 6000 platform with 150-nucleotide pair-end reads. Total reads were mapped to the Homo genome. The transcript profiling data were subjected to differential expression analysis. The RNA-seq datasets used and analysis during the current study are available from the corresponding author on reasonable request.

## Plasmids and transfection

The full-length human EGR1 cDNA fragment was cloned into a pCMV3-C-His vector (Sino Biological, China) to construct a His-tagged EGR1 eukaryotic expressing plasmid (his-EGR1, including a C-terminal His tag). The constructed plasmid was analyzed and verified by DNA sequencing. Control plasmid was provided by our laboratory. The plasmids were transfected into HeLa cells using OPTI-MEM medium and the Lipofectamine 3000 (Invitrogen, USA) reagent according to the manufacturer's protocol. EGR1 mRNA and protein abundance were measured by qPCR and western blotting respectively to examine the effect of transfection.

## Lentivirus packaging and infection

A lentiviral vector expressing EGR1 (pLV-Puro-EGR1) was used to specifically overexpress EGR1 (Sino Biological, China). The same lentiviral vector alone (pLV-Puro) was used as a negative control.

The lentiviruses were packaged by co-transfecting HEK-293T cells with the lentiviral vector plasmids and packaging plasmids, psPAX2 and pMD2.G, using PEI. Supernatants containing infectious lentiviruses were collected 48 h post-transfection. HEK-293T cells in T25 culture flasks were infected with the recombinant lentiviruses in DMEM (10% FBS) containing 2 mg/mL of puromycin for selection. The selected cell lines, resistant to puromycin, were cultured for expansion and verified by western blotting analyses and qPCR for overexpression efficacy.

## Immunofluorescence staining

The tissues were embedded and sectioned. Immunofluorescence staining was performed on the 3-µm sections. After dewaxed and hydrated, the sections were incubated primary antibodies overnight. The next day, the sections were incubated secondary antibodies. Primary antibodies included antibodies specific for EGR1, sVEGFR1 (Proteintech, Wuhan, China), and Rabbit anti-SFTSV-Gn antibodies were kindly provided by Y-Clone, Ltd. Secondary antibodies includes Alexa Fluor 488-labelled IgG or Alexa Fluor 594-labelled IgG (ThermoFisher Scientific, Wilmington, DE, USA). Cell nuclei were stained by 4′,6-diamidino-2-phenylindole (DAPI).

## Immunohistological staining

The tissues were embedded and sectioned. Immunohistochemical staining was performed on the 3-µm sections. After dewaxed and hydrated, the sections were incubated anti-EGR1 primary antibodies overnight. The next day, the sections were incubated HRP-conjugated secondary antibodies. All the sections were stained with diaminobenzidine (DAB), then counterstained with hematoxylin. The percentage of EGR1$^+$ cells was counted in a $200 \times$ field.

## Tube formation assay

In all, 96-well plates were coated with 50 µL of Matrigel (ABW, Shanghai, China) and incubated for 1 h at 37 °C. HUVECs ($1.5 \times 10^5$ cells/mL) in DMEM in the presence of the serum

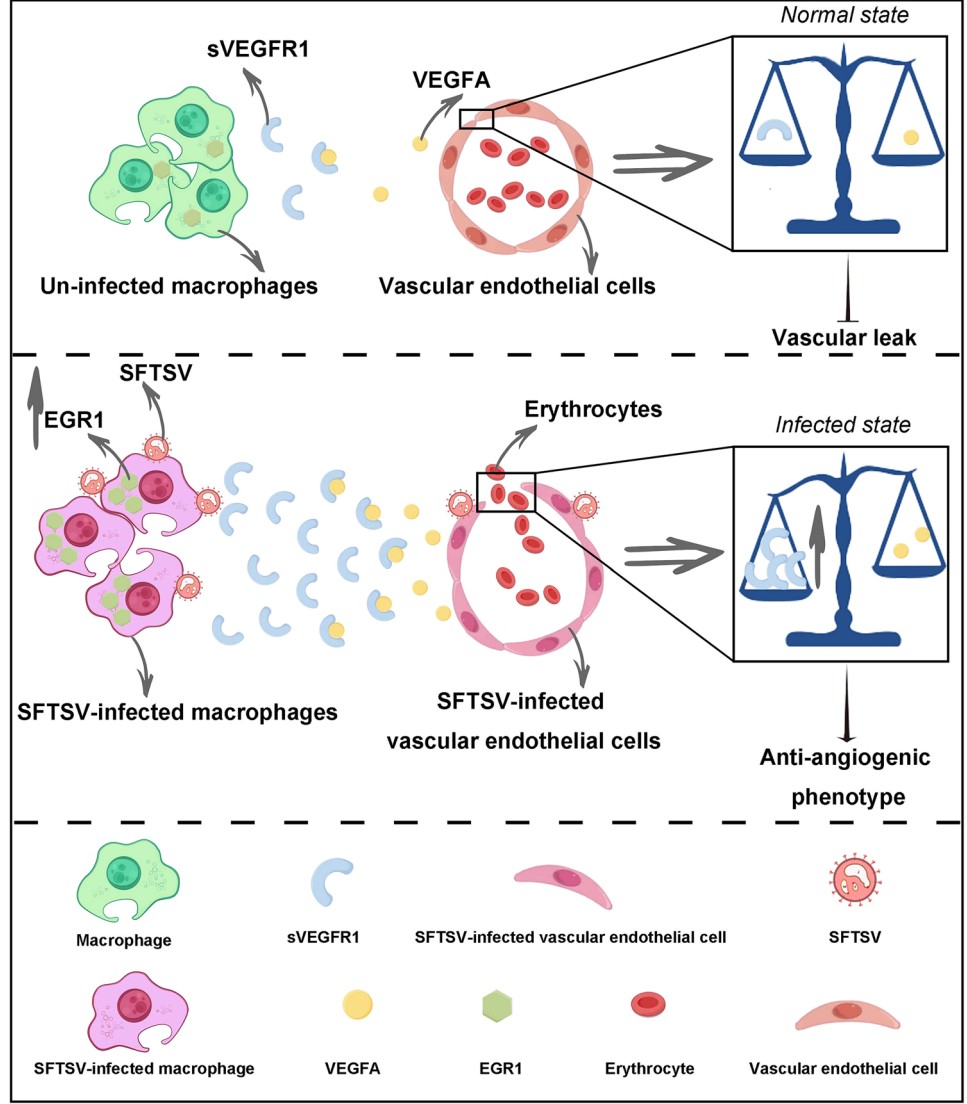

**Figure 8. Graphical illustration showing the mechanism of hemorrhage.**

The mechanism of hemorrhage that SFTSV-induced sVEGFR1 release through upregulation of EGR1 contributes to hemorrhage, suggesting that the EGR1/sVEGFR1/hemorrhage axis may be an effective target for the development of novel anti-hemorrhagic therapies during SFTSV infection.

collected and placed under UV light for 30 min for residual virus inactivation from healthy control or SFTS patients, left untreated, or treated with recombinant sVEGFR1 protein (10 ng/ml, R&D Systems, USA) alone or recombinant VEGFA protein (20 ng/ml, Proteintech, Wuhan, China) alone were seeded into 96-well plates. After the plates were incubated for 12 h or 24 h at 37 °C, the presence of tube-like structures was photographed and analyzed using a phase contrast inverted microscope. The total tube length was calculated using ImageJ software.

## Mouse aortic ring assay

The mouse aortic ring assay was performed as previously described (Baker et al, 2011). Briefly, aortas isolated from 7- to 9-week-old male C57BL/6 J were cut into ~1 mm long rings. The rings

were then placed into 96-well plates containing 100 μL of Matrigel and incubated at 37 °C in 5% $CO_2$ for 30–45 min. Culture medium supplemented with the sera of healthy control or SFTS patients was added to the wells and incubated at 37 °C. Each serum was placed under UV light for 30 min for residual virus inactivation. After 3 days or 5 days, the aortic rings were photographed using the inverted microscope. The number of microvessels emerging from the aortic rings was counted using ImageJ.

## Matrigel plug assay

The Matrigel plug assay was performed as previously described (Wang et al, 2022). Briefly, hu-PBL NCG mice were subcutaneously injected with 500 μL of Matrigel containing anti-Control Ab (10 μg/ml,

R&D Systems, USA) or anti-sVEGFR1 Ab (10 μg/ml, R&D Systems, USA). After 7 days, the Matrigel plug in hu-PBL NCG mice was harvested and photographed to determine the changes in appearance. The Matrigel plug samples were fixed with 4% paraformaldehyde for 24 h and embedded in paraffin. HE staining and IHC for CD31 were conducted to analyze angiogenesis in different groups.

## SFTSV infection of EGR1 KO mice

Heterozygous C57BL/6J mice deficient in EGR1 (EGR1$^{+/-}$; C57BL/6 JGpt-Egr1em3Cd10955in1/Gpt) were purchased from GemPharmatech (Nanjing, China). Wild-type and homozygous mutant Egr1 mice were obtained by crossing heterozygous mutant mice bearing a targeted mutation of the Egr1 gene. The deletion of EGR1 was verified by PCR assay using the primers listed in the Reagents and Tools Table. Mice were maintained on a C57BL/6J background and were age-matched in each experiment. A lethal mouse model was also established by using an anti-interferon α/β receptor subunit 1 (IFNAR1) blocking antibody (Selleck) pretreatment. Briefly, 4- to 6-week-old female EGR1$^{-/-}$ ($n = 5$) and WT ($n = 5$) C57BL/6J mice were treated with anti-IFNAR1 IgG (250 μg per mouse) by intraperitoneal injection at 1 day before and after infection. Mice were intraperitoneally inoculated with 100 μl of virus solution ($1 \times 10^5$ TCID$_{50}$) at day 0. The body weight and survival rate of the mice were monitored each day. After dissection of EGR1$^{-/-}$ and WT C57BL/6J mice at 4 dpi, the whole blood, liver, spleen, lung and kidney samples were collected for analysis.

## Hematoxylin and eosin (H&E) staining

The Matrigel plugs were fixed with 4% buffered formalin for 1 day. The fixed Matrigel plugs were embedded in paraffin, sectioned at a thickness of 3 μm, and stained with hematoxylin and eosin (H&E) to analyze angiogenesis in different groups.

## Statistics

Data analyses were performed using the GraphPad Prism 9.0 software. Results were expressed as the mean ± SD, and *n* represented the number of animals or independent experiments per group. The Student *t* test analysis was executed to compare the means of two groups, or one-way analysis of variance (ANOVA) was used for more than two groups. Statistical significance is as follows: ns, $P > 0.05$; *$P < 0.05$; **$P < 0.01$; ***$P < 0.001$.

## Study approval

The study and the protocol for this research were approved by the Center for Public Health Research, Medical School, Nanjing University. All animal experiments were approved by the Nanjing University Animal Care Committee and followed the Guide for the Care and Use of Laboratory Animals published by the Chinese National Institutes of Health. The research protocols were conducted in strict accordance and adherence to relevant policies regarding animal handling as mandated under the guidelines from the institutional animal care committee (#2014-SR-079).

## Data availability

The RNA-seq data have been deposited in the National Center for Biotechnology Information Gene Expression Omnibus database and are accessible through GEO Series accession number GSE242070.

The source data of this paper are collected in the following database record: biostudies:S-SCDT-10_1038-S44319-025-00541-2.

## Peer review information

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

## Acknowledgements

The authors thank Dr. Deyan Chen in the Center for Public Health Research, Medical School of Nanjing University for participating in experimental design. This study was supported by National Natural Science Foundation of China (U22A20335 and 31970149 to WZW).

## Author contributions

**Na Jiang**: Conceptualization; Data curation; Formal analysis; Validation; Investigation; Visualization; Writing—original draft. **Jing Wu**: Software; Formal analysis; Validation; Methodology. **Yating He**: Data curation; Software; Methodology. **Rui Zhang**: Software; Methodology. **Mengmeng Ji**: Resources; Methodology. **Linjing Zhu**: Resources; Methodology. **Shengwei Cui**: Investigation; Methodology. **Qiao You**: Software; Methodology. **Yurong Cai**: Software; Methodology. **Bingxin Liu**: Software; Methodology. **Ruining Lyu**: Methodology. **Yuxin Chen**: Resources. **Jin Zhu**: Resources; Supervision; Project administration. **Zhiwei Wu**: Conceptualization; Resources; Supervision; Funding acquisition; Project administration; Writing—review and editing.

Source data underlying figure panels in this paper may have individual authorship assigned. Where available, figure panel/source data authorship is listed in the following database record: biostudies:S-SCDT-10_1038-S44319-025-00541-2.

## Disclosure and competing interests statement

The authors declare no competing interests.

# Expanded View Figures

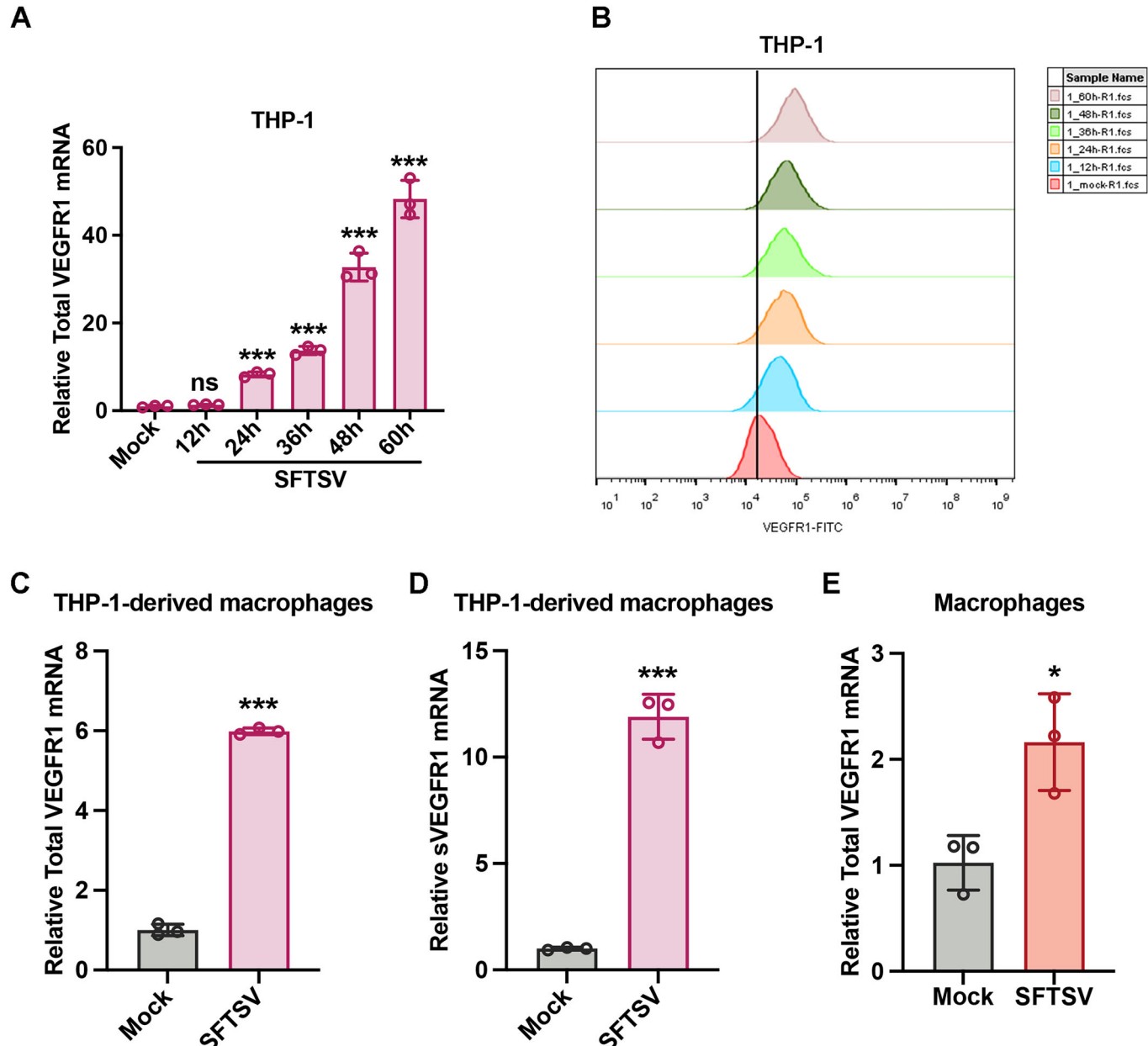

**Figure EV1. SFTSV infection up-regulates total VEGFR1 and membrane-bound VEGFR1 expression.**

(A) THP-1 cells were infected with SFTSV for 12, 24, 36, 48, and 60 h. Total VEGFR1 mRNA was measured by qPCR. $n = 3$ biological replicates. Statistical significance was determined by two-tailed unpaired Student $t$ test. (ns = 0.0548, ***$P < 0.0001$, ***$P < 0.0001$, ***$P < 0.0001$, ***$P < 0.0001$) (B) THP-1 cells were infected with SFTSV for 12, 24, 36, 48, and 60 h. Then the cells were incubated with anti-VEGFR1 for 1 h at 4 °C and VEGFR1 level on the cell membrane was determined by flow cytometry. X-axes show anti-VEGFR1 antibody (logarithm of fluorescence), Y axes depict the cell count. (C, D) THP-1 cells were induced to differentiate into macrophage-like cells by 150 nM phorbol-12-myristate-13-acetate (PMA) and then infected with SFTSV. Total VEGFR1 (C) and sVEGFR1 (D) mRNAs were measured by qPCR. $n = 3$ biological replicates. Statistical significance was determined by two-tailed unpaired Student $t$ test. ((C): ***$P < 0.0001$; (D): ***$P < 0.0001$). (E) Primary human monocyte-derived-macrophages were infected with SFTSV (MOI = 1) for 24 h. Total VEGFR1 mRNA was measured by qPCR. $n = 3$ biological replicates. Statistical significance was determined by two-tailed unpaired Student $t$ test. (*$P = 0.0198$). Data information: Data shown are mean ± SD of three biological replicates. (ns, $P > 0.05$; *$P < 0.05$; ***$P < 0.001$).

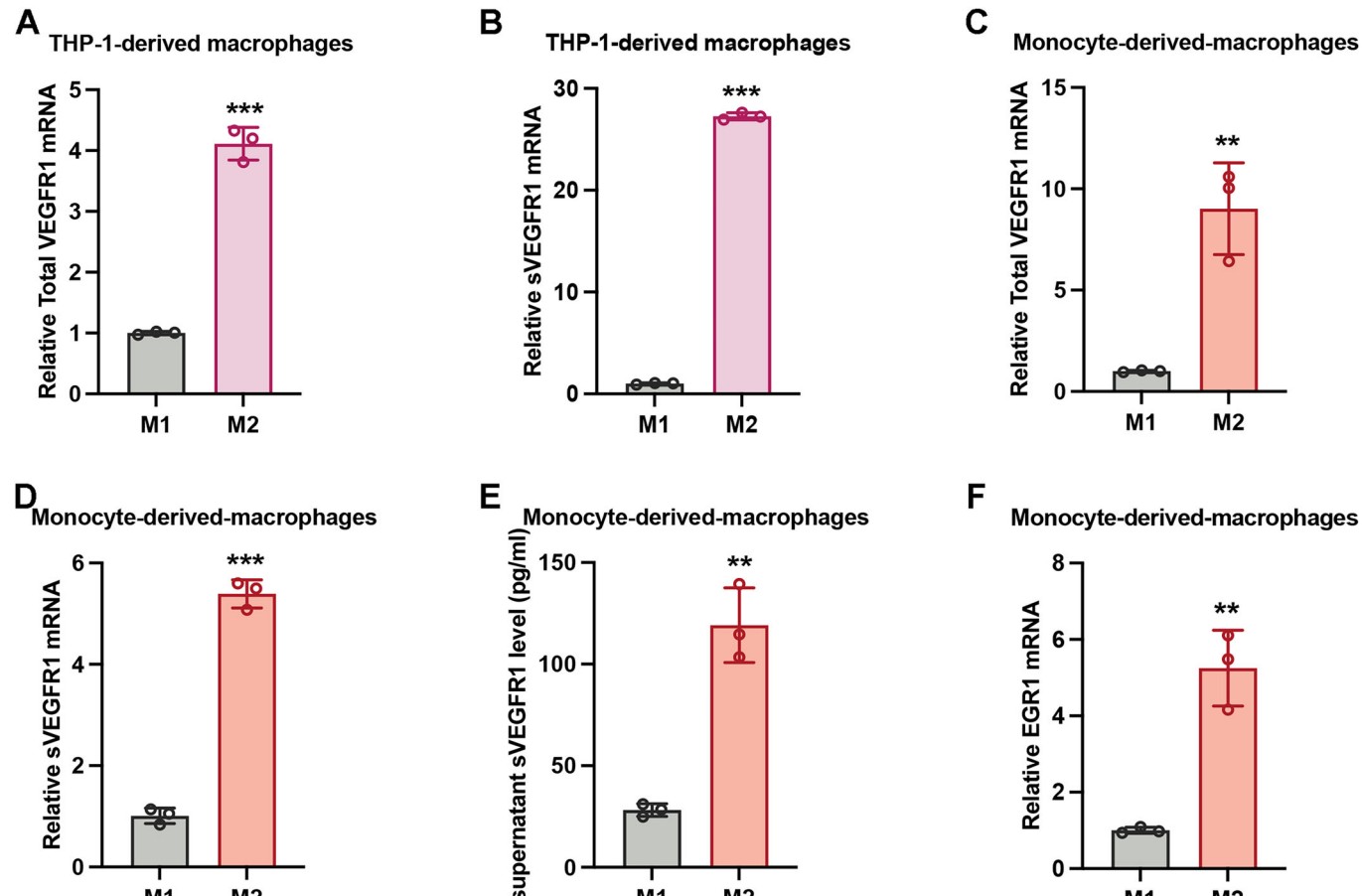

**Figure EV2.  sVEGFR1 is mainly produced by M2 macrophages.**

(A, B) THP-1 cells were induced to differentiate into M1 macrophage-like cells by 24 h incubation with 150 nM PMA, followed by IFN-γ (20 ng/ml) and LPS (10 pg/ml), or THP-1 cells were induced to differentiate into M2 macrophage-like cells by 24 h incubation with 150 nM PMA, followed by interleukin 4 (20 ng/ml) and interleukin 13 (20 ng/ml) for 72 h. Total VEGFR1 (A) and sVEGFR1 (B) mRNAs were measured by qPCR. $n = 3$ biological replicates. Statistical significance was determined by two-tailed unpaired Student $t$ test. ((A): ***$P < 0.0001$; (B): ***$P < 0.0001$). (C, D) Primary human monocyte-derived-macrophages were differentiated into M1 macrophages by culture in the presence of IFN-γ (20 ng/ml) and LPS (20 ng/ml) for 24 h, or were differentiated into M2 macrophages by culture in the presence of interleukin 4 (20 ng/ml) and interleukin 13 (20 ng/ml) for 72 h. Total VEGFR1 (C) and sVEGFR1 (D) were measured by qPCR. $n = 3$ biological replicates. Statistical significance was determined by two-tailed unpaired Student $t$ test. ((C): **$P = 0.0036$; (D): ***$P < 0.0001$). (E) Primary human monocyte-derived-macrophages were differentiated into M1 macrophages or M2 macrophages. The secreted sVEGFR1 in cell supernatant was measured by capture ELISA. $n = 3$ biological replicates. Statistical significance was determined by two-tailed unpaired Student $t$ test. (**$P = 0.0011$). (F) Primary human monocyte-derived-macrophages were differentiated into M1 macrophages or M2 macrophages. EGR1 mRNA was measured by qPCR. $n = 3$ biological replicates. Statistical significance was determined by two-tailed unpaired Student $t$ test. (**$P = 0.0018$). Data information: Data shown are mean ± SD of three biological replicates. (**$P < 0.01$; ***$P < 0.001$).

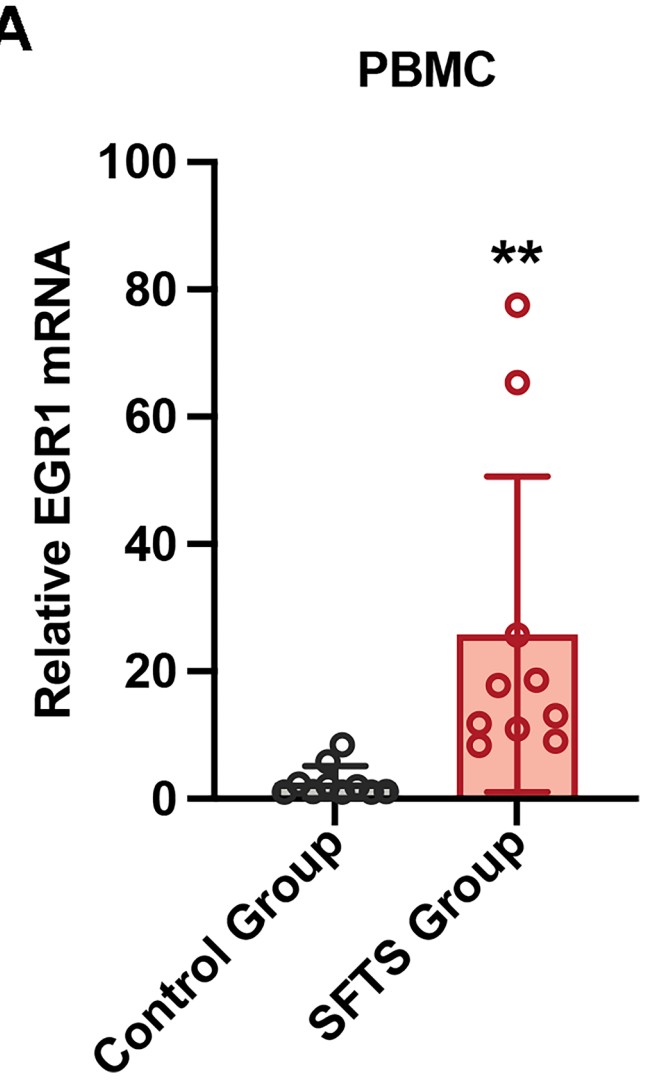

**Figure EV3.   SFTSV infection up-regulates EGR1 expression in SFTS patients.**

(**A**) PBMCs were isolated from 10 SFTS patients and 10 healthy individuals as controls. EGR1 expression in PBMCs among SFTS patients were measured by qPCR. $n = 10$ biological replicates. Statistical significance was determined by two-tailed unpaired Student $t$ test. (**P = 0.0085). Data information: Data shown are mean ± SD of ten biological replicates with each data point representing a biological experiment (**P < 0.01).

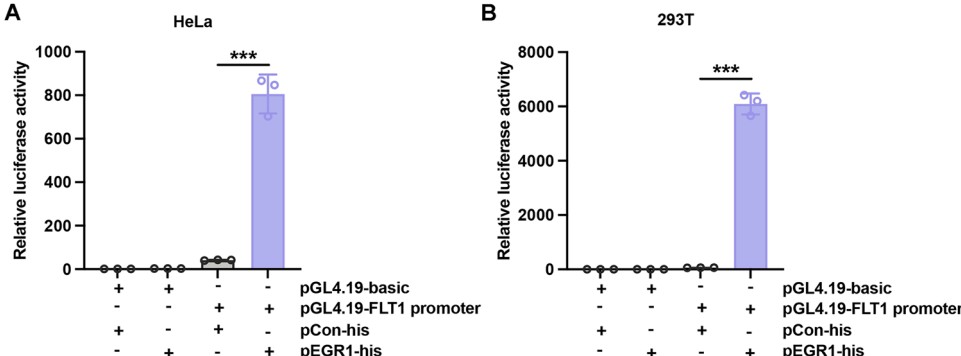

**Figure EV4. Overexpression of EGR1 activates the VEGFR1 promoter.**

(A, B) Luciferase reporter assay. For the VEGFR1 promoter luciferase reporter assay, the predicted binding sites of VEGFR1 and EGR1 were obtained from the eukaryotic promoter database (EPD). The corresponding promoter constructs in the promoter region of VEGFR1 was synthesized and then inserted into the pGL4.19-basic firefly luciferase reporter vector (PPL), named pGL4.19-FLT1 promoter. These vectors were co-transfected into HeLa or HEK-293T cells with EGR1 construct or empty vector. Measurement of luciferase activity was conducted at 48 h post-transfection. Representative results were from three independent experiments. $n = 3$ biological replicates. Statistical significance was determined by two-tailed unpaired Student $t$ test. ((A): \*\*\*$P = 0.0001$; (B): \*\*\*$P < 0.0001$). Data information: Data shown are mean ± SD of three biological replicates (\*\*\*$P < 0.001$).

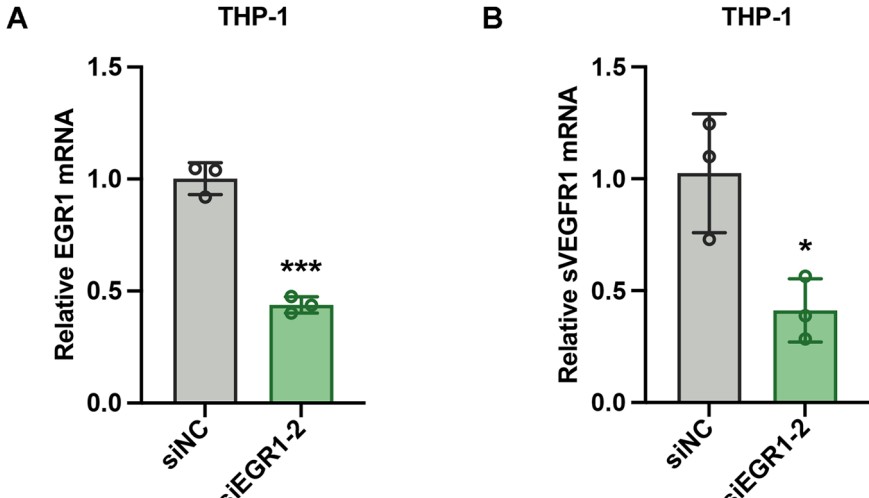

**Figure EV5. Knock-down of EGR1 reduces sVEGFR1 expression.**

(A, B) THP-1 cells transfected with siNC or siEGR1 (siEGR1-2 in Fig. 5G) were harvested at 36 hpt to evaluate EGR1 mRNA (A) knock-down efficiency and sVEGFR1 mRNA (B) levels by qPCR. $n = 3$ biological replicates. Statistical significance was determined by two-tailed unpaired Student $t$ test. ((A): ***$P = 0.0003$; (B): *$P = 0.0243$). Data information: Data shown are mean ± SD of three biological replicates (*$P < 0.05$; ***$P < 0.001$).

