## [Peer Review File · EMBO Reports]

sVEGFR1 up-regulation via EGR1 impairs vascular repair in SFTSV-induced hemorrhage

Zhiwei Wu, Na Jiang, Jing Wu, Yating He, Rui Zhang, mengmeng Ji, Linjing Zhu, Shengwei Cui, Qiao You, Yurong Cai, Bingxin Liu, Ruining Lyu, YuXin Chen, and Jin Zhu

Corresponding author(s): Zhiwei Wu (wzhw@nju.edu.cn) , Jin Zhu (zhujin1968@njmu.edu.cn)

Review Timeline:

Submission Date:	17th Feb 25
Editorial Decision:	25th Mar 25
Revision Received:	5th Jun 25
Editorial Decision:	3rd Jul 25
Revision Received:	10th Jul 25
Accepted:	20th Jul 25

Editor: Achim Breiling

Transaction Report:

Dear Dr. Wu,

Thank you for the submission of your manuscript to EMBO reports. I have now received the reports from the three referees that were asked to evaluate your study, which can be found at the end of this email.

As you will see, the referees think that these findings are of interest. However, they have several comments, concerns, and suggestions, indicating that a major revision of the manuscript is necessary to allow publication of the study in EMBO reports. As the reports are below, and all the referee concerns need to be addressed, I will not detail them here. Nevertheless, I consider it important that the discrepancies regarding the cell lines used (mentioned by all three referees) are sorted out and essential experiments have been performed/replicated in primary human monocytes or in vitro differentiated primary human monocytes.

Given the constructive referee comments, I would like to invite you to revise your manuscript with the understanding that the concerns of the referees must be addressed in the revised manuscript and in a detailed point-by-point response. Acceptance of your manuscript will depend on a positive outcome of a second round of review. It is EMBO reports policy to allow a single round of revision only and acceptance of the manuscript will therefore depend on the completeness of your responses included in the next, final version of the manuscript.

- 1) a .docx formatted version of the final manuscript text (including legends for main figures, EV figures and tables), but without the figures included. Figure legends should be compiled at the end of the manuscript text.
- 2) individual production quality figure files as .eps, .tif, .jpg (one file per figure), of main figures and EV figures. Please upload these as separate, individual files upon re-submission.

- 4) a complete author checklist, which you can download from our author guidelines

(<https://www.embopress.org/page/journal/14693178/authorguide>). Please insert page numbers in the checklist to indicate where the requested information can be found in the manuscript. The completed author checklist will also be part of the RPF.

5) that primary datasets produced in this study (e.g. RNA-seq, CHIP-seq, structural and array data) are deposited in an appropriate public database. If no primary datasets have been deposited, please also state this in a dedicated section (e.g. 'No primary datasets have been generated and deposited'), see below.

The accession numbers and database should be listed in a formal "Data Availability" section that follows the model below. This is now mandatory (like the COI statement). Please note that the Data Availability Section is restricted to new primary data that are part of this study. This section is mandatory. As indicated above, if no primary datasets have been deposited, please state this in this section

Data availability

8) Regarding data quantification and statistics, please make sure that the number "n" for how many independent experiments were performed, their nature (biological versus technical replicates), the bars and error bars (e.g. SEM, SD) and the test used to calculate p-values is indicated in the respective figure legends (also for EV and Appendix figures). Please also check that all the p-values are explained in the legend, and that these fit to those shown in the figure. Please provide statistical testing where applicable. Please avoid the phrase 'independent experiment', but clearly state if these were biological or technical replicates. Please also indicate (e.g. with n.s.) if testing was performed, but the differences are not significant. In case n=2, please show the data as separate datapoints without error bars and statistics. See also: <http://www.embopress.org/page/journal/14693178/authorguide#statisticalanalysis>

9) Please add scale bars of similar style and thickness to microscopic images, using clearly visible black or white bars (depending on the background). Please place these in the lower right corner of the images themselves. Please do not write on or near the bars in the image but define the size in the respective figure legend.

10) Please also note our reference format:

12) We now use CRediT to specify the contributions of each author in the journal submission system. CRediT replaces the author contribution section. Please use the free text box to provide more detailed descriptions and do NOT provide your final manuscript text file with an author contributions section. See also our guide to authors: <https://www.embopress.org/page/journal/14693178/authorguide#authorshipguidelines>

13) All Materials and Methods need to be described in the main text using our 'Structured Methods' format, which is required for all research articles. According to this format, the Methods section should include a Reagents and Tools Table (listing key reagents, experimental models, software, and relevant equipment and including their sources and relevant identifiers), uploaded as separate file, and a Methods section in which we encourage the authors to describe their methods using a step-by-step protocol format with bullet points, to facilitate the adoption of the methodologies across labs. More information on how to adhere to this format as well as downloadable templates (.doc) for the Reagents and Tools Table can be found in our author guidelines (section 'Structured Methods'):

14) Please order the sections like this, using these names:

Title page - Abstract - Keywords - Introduction - Results - Discussion - Methods - Data availability section - Acknowledgements (including the funding information) - Disclosure and Competing Interests Statement - References - Figure legends - Expanded View Figure legends

15) Please make sure that all the funding information is also entered into the online submission system and that it is complete and similar to the one in the acknowledgement section of the manuscript text file.

I look forward to seeing a revised form of your manuscript when it is ready.

Yours sincerely,

Referee #1:

The manuscript by Jiang et al entitled SFTSV causes hemorrhage by up-regulating sVEGFR1 through inducing EGR1 and impairing vascular repair describes the endothelial effect of severe fever with thrombocytopenia syndrome virus (SFTSV) infection. The authors used patient samples, mouse models, HUVECs and cell lines to show that SFTSV infection increased soluble vascular endothelial growth factor-receptor 1 and that this inhibits angiogenesis. In addition, the manuscript shows that this increase is mediated by early growth response gene 1. The authors concluded that increases in soluble vascular endothelial growth factor-receptor 1 contribute to haemorrhage by disrupting the angiogenic balance and interfering with endothelial repair.

The manuscript represents a substantial amount of work and is overall well-presented and clear. This is an important finding that points towards a target with therapeutic potential to address haemorrhage in SFTSV infections.

Comments:

Some aspects of the manuscript require changes/clarifications before publication.

Please define terms when first used (eg, sVEGFR1 in the introduction).

Lines 140-141 state "Collectively, these results unequivocally confirmed that SFTSV infection triggered the release of sVEGFR1 from monocytes/macrophages"

Please change to "Collectively, these results unequivocally confirmed that SFTSV infection triggered the release of sVEGFR1 from THP-1 cells"

Methods. The methods section indicates that mice were infected with SFTSV and then "Spleens and blood cells were collected immediately after animal sacrifice". Please clarify this sentence: does it mean immediately after virus injection or immediately after culling the animals? I assume it's the second option, in that case, how long were the animals kept after infection with the virus?

Mock treatment: please describe how mock-treatment was carried out

Line 450 states "PBMC engraftments were evaluated at the 21st day after the transplantation by FACS." No data was presented as to the levels of engraftment. Was engraftment comparable in all mice? Do different levels of engraftment affect expression of sVEGFR1 in different mice as shown in Fig 2b? Is that the case, why were only 5 mice used in these experiments?

Fig 3c. Even though there was an increase in tube formation after removal of sVEGFR1 from patient sera, the increase is quite modest. Did the authors ascertain that all sVEGFR1 had been removed by the magnetic beads? Or are there other factors not shown in the paper likely responsible for tube formation inhibition? Why was sera from only 3 patients and controls used in these experiments?

HUVECs. Please describe the HUVECs cells used in this paper (obtained locally or from commercial sources, passage number of the cells used in these assays, any growth factors in the culture media?)

Fig. 3d shows decrease in tube formation upon addition of exogenous sVEGFR1 to cells treated with control sera. Would something similar be observed in the absence of human serum? (ie, just adding sVEGFR1 to Matrigel?)

Fig. 5. Consistency. The text mentions HeLa cell transfection with pEGR1-his or control plasmid pCon-his. The Figure is labelled EGR1-his and EV-his. This can be confusing (please use either Con-his or EV-his).

Fig 5d and g. The Western blot in Fig 5d shows very low levels of both EGR1 and sVEGFR1 in 293T cells, however the blot in Fig 5g shows high expression of these two proteins in the same cells. Is this a matter of membrane exposure? Or protein loading? Please clarify.

The introduction states that monocytes/macrophages are the main cellular target of SFTSV (Zhang et al, 2019) and for this reason monocyte-like THP-1 cells were used as an infection model. Fig. 5 shows other cell lines (HeLa, HEK293-T) that may be less relevant in the mechanism of SFTSV infection. Were these cell lines chosen for practical reasons?

THP-1 cells have been used before in siRNA knockdown studies, why were these cells not used in the current study to knockdown EGR1? CRISPR knockout in THP-1 cells might have been more appropriate in this context.

Fig 7i. The bands in this blot (especially GAPDH) are distorted, which makes quantification inadequate. Could this blot be improved?

Discussion. The discussion seems too long. Please shorten the discussion to address the key points of the paper. A substantial part of the discussion talks about supplementary data that was not presented in the main results section. Some of these results could be included in the results section.

Referee #2:

The study by Na et al reports on a lesser known disease SFTS where authors report that SFTSV induced sVEGFR1 up-regulation via early growth response gene (EGR1), of which VEGFR1 is a downstream target. Some comments for consideration.

1. The use of ThP1 cell line is flawed as it is well reported that it is not the same as primary macrophages, and their behavior may not be identical.
2. Have the authors done a comparison in simulating the age of the animals to mirror real like human infections where age is a factor for disease severity?
3. There are available data that depicts WNV severity and hemorrhagic fever and overlapping disease manifestations. Author should do a meta-analysis on these datasets.

Referee #3:

The authors show in this study that SFTSV infection leads to an upregulation of the EGR1-sVEGFR1 axis, with a consequential impairment/reduction of angiogenesis both in vitro and in vivo. sVEGFR1 is upregulated in an EGR1-dependent manner and seems one of the main reasons of vascular leakage and hence haemorrhage in addition to thrombocytopenia in haemorrhagic

fever virus infection. By using knock-out models, pathogenesis can be ameliorated as well as depletion of sVEGFR1 or targeting sVEGFR1 via an antibody indicated that the mechanism could be therapeutically exploited. The experiments are comprehensive and overall the conclusions are supported by the data. In general, the study would benefit from confirming key findings in the relevant model of primary human macrophages (-derived from monocytes).

I have the following comments:

- It is of interest that sVEGFR1 is more abundantly upregulated in M2-like macrophages and that the EGR1-KO attenuated viral replication, hinting that either EGR1 or its downstream effectors are potentially viral dependency factors. However the enthusiasm related to these results is largely limited by the usage of differentiated THP1 cells for this experiment. It is straightforward to differentiate M1/M2 macrophages from primary human monocytes, so I wonder why the authors, even though corroborating their results with primary cells throughout, fail to do so here.
- In Fig. 1A, the qPCR was done up to 60 h.p.t. and shows a nearly 300x increase in sVEGFR1 while the latest WB timepoint is at 48 h.p.t.. Do the authors have any reason for not processing the 60 h.p.t. for WB as well?
- Fig.1: How do the authors justify that they had more vRNA at 36 h.p.i. (see panel F) compared to a later time point (see panel G)?
- Line 141: Again here and throughout, be precise in description of the model used; THP1 cells are not monocytes/macrophages; when differentiated they are macrophage-like and without differentiation a monocytic cell line.
- Figure 3: It is unclear to me how was this quantified. Did the authors quantify the number of tubes or the area?
- Line 183: "significant enhancement of tube formation" This is not shown in Fig. 3D, as the only quantification plotted here is the normalized comparison between healthy control group and the one with the added sVEGFR1.
- Figure 4: It is important to know which other genes were significantly upregulated, as there are clearly some that are more upregulated than EGR1 in the dot plot.
- Fig.4: Is EGR1 also upregulated in the serum derived from infected patients?
- Fig.5: It would make more sense to swap panels A and B, to be consistent with all previous figures, showing first the mRNA and then the protein data.
- Fig.5g: Why are the statistical tests performed between siRNA 1 and siRNA3, instead of the ctrl siRNA?
- Fig.7e: Is the AUC significantly different between the Ctrl and the EGR1-KO mice?
- Lines 273-275: also suggests that EGR1 KO mice are less susceptible to SFTSV. Does the viral load correlate with the lower NP levels?
- Lines 306-307: the authors should add that this was observed in THP-1 cells. At first glance, I thought this data came from BMDMs
- Lines 317-318: would be important to confirm in primary human monocyte-derived-macrophages

Dear Dr. Breiling,

I thank you very much for your time in editing the manuscript and for giving us the opportunity to revise. I also greatly appreciate the reviewers for their meticulous reading of the manuscript and raise many excellent questions. We have carefully gone through the comments and suggestions, performed additional experiments, addressed the reviewers' concerns and revised the manuscript accordingly. The revised manuscript, which has been re-submitted along with this letter, is a much-improved version and will, I wish, satisfy the rigor of the reviewers and the high standard of the journal. The point-to-point answer to the reviewers' questions is attached at the end of this letter. All the changes are highlighted in the revised manuscript. Please do not hesitate to contact me if you or the reviewers have additional questions. I am

Truly yours

Allen Z. Wu, Ph.D.
Professor and Director
Center for Public Health Research
Medical School, Nanjing University
Professor and Associate Director
State Key Laboratory of Analytical Chemistry for Life Science
Nanjing University
22 Hankou Road, Nanjing, Jiangsu 210093. China

Editor:

Comments to the Author

Thank you for the submission of your manuscript to EMBO reports. I have now received the reports from the three referees that were asked to evaluate your study, which can be found at the end of this email.

As you will see, the referees think that these findings are of interest. However, they have several comments, concerns, and suggestions, indicating that a major revision of the manuscript is necessary to allow publication of the study in EMBO reports. As the reports are below, and all the referee concerns need to be addressed, I will not detail them here. Nevertheless, I consider it important that the discrepancies regarding the cell lines used (mentioned by all three referees) are sorted out and essential experiments have been performed/replicated in primary human monocytes or in vitro differentiated primary human monocytes.

Given the constructive referee comments, I would like to invite you to revise your manuscript with the understanding that the concerns of the referees must be addressed in the revised manuscript and in a detailed point-by-point response. Acceptance of your manuscript will depend on a positive outcome of a second round of review. It is EMBO reports policy to allow a single round of revision only and acceptance of the manuscript will therefore depend on the completeness of your responses included in the next, final version of the manuscript.

A: I appreciate your efforts in editing the manuscript and the reviewers' many valuable comments and suggestions. We have considered all the comments from the editor and reviewers, performed additional experiments and revised the manuscript accordingly.

Referee #1:

Comments to the Author

The manuscript by Jiang et al entitled SFTSV causes hemorrhage by up-regulating sVEGFR1 through inducing EGR1 and impairing vascular repair describes the endothelial effect of severe fever with thrombocytopenia syndrome virus (SFTSV) infection. The authors used patient samples, mouse models, HUVECs and cell lines to show that SFTSV infection increased soluble vascular endothelial growth factor-receptor 1 and that this inhibits angiogenesis. In addition, the manuscript shows that this increase is mediated by early growth response gene 1. The authors concluded that increases in soluble vascular endothelial growth factor-receptor 1 contribute to haemorrhage by disrupting the angiogenic balance and interfering with endothelial repair.

The manuscript represents a substantial amount of work and is overall well-presented and clear. This is an important finding that points towards a target with therapeutic potential to address haemorrhage in SFTSV infections.

Comments:

Some aspects of the manuscript require changes/clarifications before publication.

A: We would like to thank the reviewer for his/her efforts in pointing out the shortcomings of the manuscript. We have considered all comments, performed additional experiments and revised the manuscript to address the reviewer's concerns.

1. Please define terms when first used (eg, sVEGFR1 in the introduction).

A1: The reviewer's suggestion is well taken and we have provided definitions for terms that were used for first time in the revised manuscript, and the changes are highlighted in yellow (**Lines 29-31 and Lines 97-98**).

2. Lines 140-141 state "Collectively, these results unequivocally confirmed that SFTSV infection triggered the release of sVEGFR1 from monocytes/macrophages". Please change to "Collectively, these results unequivocally confirmed that SFTSV infection triggered the release of sVEGFR1 from THP-1 cells"

A2: The reviewer's suggestion is well taken and we have changed the sentence as suggested in the revised manuscript. The change was highlighted in yellow (**Lines 140-141**).

3. Methods. The methods section indicates that mice were infected with SFTSV and then "Spleens and blood cells were collected immediately after animal sacrifice". Please clarify this sentence: does it mean immediately after virus injection or immediately after culling the animals? I assume it's the second option, in that case, how long were the animals kept after infection with the virus?

Mock treatment: please described how mock-treatment was carried out

A3: Apology for the confusion. The animals were infected for 14 days before being sacrificed and organs being taken for examination. These sentences have been modified as highlighted in yellow in **Lines 485-488** as following:

Hu-PBL mice (n = 5) were injected intraperitoneally with SFTSV (1×10^3 TCID₅₀) and mock-infected mice (n = 5) were injected intraperitoneally with the same volume of DMEM as controls. Spleen and blood cells were collected at 14 days post infection at the time of sacrifice.

4. Line 450 states "PBMC engraftments were evaluated at the 21st day after the transplantation by FACS." No data was presented as to the levels of engraftment. Was engraftment comparable in all mice? Do different levels of engraftment affect expression of sVEGFR1 in different mice as shown in Fig 2b? Is that was the case, why were only 5 mice used in these experiments?

A4: Thanks for these excellent questions from the reviewer. We have provided engraftment data in **Appendix Figure S5**. The data and the experimental descriptions as being presented below:

According to our previously reported method (Ji *et al*, 2024; Xu *et al*, 2021) in the construction of Hu-PBL NCG mouse model, blood was taken from the mice at the 21st day after the human PBL engraftment to evaluate the percentage of human CD45 by FACS. The mice with over 5% human CD45 in PBMC was considered as

successfully humanized. To minimize variability, only mice with comparable engraftment levels were included and randomly divided into uninfected control (n = 5) or infected (n = 5) groups. Our results showed no significant variations at the levels of engraftment in all mice between two groups.

Appendix Figure S5

Appendix Figure S5. The levels of engraftment of hu-PBL NCG.

(A-B) Flow cytometry analysis of the percentage of human CD45⁺ cells. n = 5 biological replicates. Statistical significance was determined by two-tailed unpaired *Student t*-test. (ns = 0.5664)

Data information: Data shown were *Mean* ± *SD* of five biological replicates. (ns, *P* > 0.05)

Due to the high cost and technical complexity of generating humanized mice, we limited the cohort to 5 animals per group. Despite the limited sample size, our engraftment showed that the engraftment was consistent and yielded minimal variations among engrafted mice. In addition, significant differences in sVEGFR1 levels ensured the reliability of the findings.

5. Fig 3c. Even though there was an increase in tube formation after removal of sVEGFR1 from patient sera, the increase is quite modest. Did the authors ascertain that all sVEGFR1 had been removed by the magnetic beads? Or are there other factors not shown in the paper likely responsible for tube formation inhibition? Why was sera from only 3 patients and controls used in these experiments?

A5: We thank the reviewer for these excellent questions. Regarding the first question, we used sVEGFR1-specific magnetic beads for depletion and verified the efficiency of sVEGFR1 removal by ELISA. The results confirmed a substantial reduction in sVEGFR1 levels in the sera after treatment. However, we acknowledge that complete depletion could not be guaranteed, likely due to limited antibody availability and binding capacity of the beads. We have provided the data in **Appendix Figure S6** in the revised manuscript. The data and the experimental descriptions as being presented below:

Sera from healthy individuals and SFTS patients were placed under UV light for 30 min to inactivate the virus, and then pretreated with a sVEGFR1 neutralizing antibody-conjugated or control antibody-conjugated magnetic beads to remove sVEGFR1. The efficiency of sVEGFR1 removal were verified by ELISA and the results showed a substantial reduction in sVEGFR1 levels in the sera after sVEGFR1 antibody treatment.

Appendix Figure S6

Appendix Figure S6. The efficiency of sVEGFR1 removal.

(A) Sera from healthy individuals and SFTS patients were placed under UV light for 30 min to inactivate the virus, and then pretreated with a sVEGFR1 neutralizing antibody-conjugated or control antibody-conjugated magnetic beads to remove sVEGFR1. sVEGFR1 in sera were measured by capture ELISA. ND, Not Detected. n = 2 technical replicates.

As for the modest increase in tube formation observed after sVEGFR1 removal, we agree that additional factors beyond sVEGFR1 may contribute to the inhibitory effects on endothelial function. SFTSV-infected immune cells, including low-density neutrophils, mast cells, monocytes, and macrophages, could release vasoactive mediators such as chymase and tryptase, which disrupt endothelial tight junctions and compromise microvascular integrity (Li *et al.*, 2017; Li *et al.*, 2019; Wang *et al.*, 2022b; Xu *et al.*, 2021). Furthermore, these cells secrete elevated levels of proinflammatory cytokines (e.g., TNF- α , IL-1 β , and IL-6), which can destabilize endothelial junctions, and induce endothelial apoptosis (Xu *et al.*, 2021). These additional factors likely contribute to the residual inhibitory effect observed even after sVEGFR1 depletion. Finally, we used sera from only three patients and three controls in this specific assay due to the fact that the infection is opportunistic and rarely many SFTS patients were available at the same time. To further support our observations, we conducted an additional experiment using sera from three additional patients and three healthy controls. The results were consistent with our initial findings and are now presented in **Appendix Figure S7** in the revised manuscript, as described below:

Appendix Figure S7

Appendix Figure S7.

(A) Sera from healthy individuals and SFTS patients were placed under UV light for 30 min to inactivate the virus, and then pretreated with a sVEGFR1 neutralizing antibody-conjugated or

control antibody-conjugated magnetic beads to remove sVEGFR1. HUVECs were introduced into the Matrigel in the presence of pretreated-serum from either healthy individuals or SFTS patients, and tube formation of HUVECs was examined at 12 h after treatment. Representative capillary tubule structures were shown. Scale bar = 50 μ m. Bar graph on the right represent the fold change of tubule formation. n = 3 biological replicates. Statistical significance was determined by two-tailed unpaired *Student t*-test. (Control: * P = 0.0342; SFTS: *** P = 0.0010)

Data information: Data shown were *Mean* \pm *SD* of three biological replicates. (*, P < 0.05; ***, P < 0.001)

6. HUVECs. Please describe the HUVECs cells used in this paper (obtained locally or from commercial sources, passage number of the cells used in these assays, any growth factors in the culture media?)

A6: Thanks for your comments. We have added the respective details on the HUVECs cells used in this paper and highlighted the changes in yellow in **Lines 468-471** in the revised manuscript as following:

HUVECs were obtained from the American Type Culture Collection (ATCC, USA). HUVECs (Passages 3-6) were acclimatized for 1-2 days either in working media (DMEM with 10% FBS) or starving media (DMEM without FBS) before the experimental start.

Tube formation assay: HUVECs (Passages 3-6) were cultured in starving media.

7. Fig. 3d shows decrease in tube formation upon addition of exogenous sVEGFR1 to cells treated with control sera. Would something similar be observed in the absence of human serum? (ie, just adding sVEGFR1 to Matrigel?)

A7: Thank you for the insightful question. We conducted additionally experiment where exogenous sVEGFR1 was added directly to the Matrigel in the absence of human serum and the results were presented in **Appendix Figure S8** in the revised manuscript as described below:

HUVECs were introduced into the Matrigel in the absence of human serum, following addition with or without exogenous recombinant sVEGFR1 (10 ng/ml). Tube

formation of HUVECs was examined at 12 h after treatment. The results showed no significant inhibition of tube formation, indicating that additional factors present in the human serum are likely required to mediate the anti-angiogenic effect observed in our study.

Appendix Figure S8

Appendix Figure S8.

(A) HUVECs were introduced into the Matrigel in the absence of human serum, following addition with or without exogenous recombinant sVEGFR1 (10 ng/ml). Tube formation of HUVECs was examined at 12 h after treatment. Scale bar = 50 μ m. Bar graph on the right represent the fold change of tubule formation. n = 3 biological replicates. Statistical significance was determined by two-tailed unpaired *Student t*-test. (ns = 0.5944)

Data information: Data shown were *Mean* \pm *SD* of three biological replicates. (ns, $P > 0.05$)

8. Fig. 5. Consistency. The text mentions HeLa cell transfection with pEGR1-his or control plasmid pCon-his. The Figure is labelled EGR1-his and EV-his. This can be confusing (please used either Con-his or EV-his).

A8: Apology for the confusion. We have corrected it in the revised manuscript and the change was highlighted in yellow (**Lines 249-251**) as following:

To explore the effect of increased EGR1 expression on sVEGFR1 induction, HeLa cells were transiently transfected with a plasmid expressing EGR1 (EGR1-his) or an empty vector (EV-his).

9. Fig 5d and g. The Western blot in Fig 5d shows very low levels of both EGR1 and sVEGFR1 in 293T cells, however the blot in Fig 5g shows high expression of these two proteins in the same cells. Is this a matter of membrane exposure? Or protein loading? Please clarify.

A9: The observed differences in EGR1 and sVEGFR1 band intensities between **Fig. 5D** (now it is **Fig. 5E** in the revised manuscript) and **Fig. 5G** are primarily due to differences in protein loading and exposure settings during Western blot analysis. In **Fig. 5D** (**Fig. 5E** in the revised manuscript), EGR1 was overexpressed, resulting in higher protein levels; therefore, we intentionally reduced the amount of protein loaded to avoid signal oversaturation. Simultaneously, in **Fig. 5D** (**Fig. 5E** in the revised manuscript), we adjusted the exposure time to better visualize the EGR1 signal. Importantly, GAPDH was used as the internal loading control in both panels. The different intensity of GAPDH bands across lanes confirms that the observed differences in EGR1 and sVEGFR1 levels reflect experimental adjustments in protein loading and exposure.

10. The introduction states that monocytes/macrophages are the main cellular target of SFTSV (Zhang et al, 2019) and for this reason monocyte-like THP-1 cells were used as an infection model. Fig. 5 shows other cell lines (HeLa, HEK293-T) that may be less relevant in the mechanism of SFTSV infection. Were these cell lines chosen for practical reasons?

THP-1 cells have been used before in siRNA knockdown studies, why were these cells not used in the current study to knockdown EGR1? CRISPR knockout in THP-1 cells might have been more appropriate in this context.

A10: We appreciate the reviewer's comments. HeLa and HEK293T cells were employed in our study primarily due to their high transfection efficiency and suitability for mechanistic studies involving promoter activity and overexpression.

Regarding EGR1 knockdown, while we recognize that THP-1 cells are a more physiologically relevant model, they are technically challenging for gene editing.

However, to address reviewer's concerns, we conducted additional experiment using siRNA and showed that knock-down of EGR1 with siRNA in THP-1 cells by transfecting the cells with EGR1 siRNA significantly reduced the sVEGFR1 expression as shown by qPCR (**Appendix Figure S1**), further demonstrating the important role of EGR1 in promoting sVEGFR1 expression.

The data and the experimental description were added to the revised manuscript as presented below:

Appendix Figure S1

Appendix Figure S1.

(**A-B**) THP-1 cells transfected with siNC or siEGR1 (siEGR1-2 in Fig. 5G) were harvested at 36 hpt to evaluate EGR1 mRNA (**A**) knock-down efficiency and sVEGFR1 mRNA (**B**) levels by qPCR. $n = 3$ biological replicates. Statistical significance was determined by two-tailed unpaired *Student t*-test. (A: *** $P = 0.0003$; B: * $P = 0.0243$)

Data information: Data shown were *Mean* \pm *SD* of three biological replicates. (*, $P < 0.05$; ***, $P < 0.001$)

We also performed CRISPR/Cas9-engineered knockout of EGR-1 in THP-1 cells, which necessitated lentiviral transduction for efficient delivery of the Cas9 and gRNA constructs. However, we found that THP-1 cells which were normally maintained in suspension exhibited partial adherence following lentiviral transduction. This phenotypic shift affects cellular behavior, including cell proliferation and subsequent

monoclonal selection, thereby complicating downstream analysis. As a result, we employed siRNA-mediated knockdown, which preserved the typical suspension phenotype and ensured more consistent data.

11. Fig 7i. The bands in this blot (especially GAPDH) are distorted, which makes quantification inadequate. Could this blot be improved?

A11: The reviewer's suggestion is well taken. We have repeated the Western blot experiment using fresh samples under optimized running conditions, as presented in **Figure 7I**.

Images were presented below and the experimental descriptions were as follows:

sVEGFR1 and the viral NP protein in spleens were measured by Western blotting (left). The sVEGFR1 and SFTSV-NP protein expression relative to GAPDH was shown (right), respectively. These results showed that the protein level of sVEGFR1 and SFTSV-NP in spleens were both decreased in the KO mice.

Figure 7

12. Discussion. The discussion seems too long. Please shorten the discussion to address the key points of the paper. A substantial part of the discussion talks about supplementary data that was not presented in the main results section. Some of these results could be included in the results section.

A12: The reviewer's suggestions are well taken. We have shortened the discussion section to better address the key points of the paper and integrated supplementary data previously only mentioned in the discussion section into the results section. The

revised text was highlighted in yellow in the revised manuscript (**Lines 143-160, Lines 259-263 and Lines 399-403**).

Referee #2:

Comments to the Author

The study by Na et al reports on a lesser known disease SFTS where authors report that SFTSV induced sVEGFR1 up-regulation via early growth response gene (EGR1), of which VEGFR1 is a downstream target. Some comments for consideration.

1. The use of ThP1 cell line is flawed as it is well reported that it is not the same as primary macrophages, and their behavior may not be identical.

A1: The reviewer's suggestions are well taken. We conducted additionally experiments using primary human monocyte-derived macrophages to validate our findings. The images were presented below and the experimental descriptions were as follows:

CD14⁺ monocytes were isolated from PBMCs via positive selection using CD14 microbeads and subsequently differentiated into macrophages over 7 days in the presence of recombinant human M-CSF. These macrophages were then infected with SFTSV for 24 h. sVEGFR1 expression in macrophages was measured and the results were combined with the data of **Fig. S1** of the original manuscript and now labelled as **Fig. EV2A-E** and the original **1A-C** became **2F-H** in the revised manuscript. The results showed that the mRNA level of sVEGFR1 and total VEGFR1 were increased in the infected macrophages (**Fig. EV2A-B, D in the revised manuscript**). To further assess the capability of SFTSV inducing sVEGFR1 secretion, we performed an ELISA and demonstrated that SFTSV-infected macrophages produced significantly higher sVEGFR1 in the supernatants as compared with that of the uninfected macrophages (**Fig. EV2C in the revised manuscript**). Collectively, these results unequivocally confirmed that SFTSV infection triggered the release of sVEGFR1 from primary macrophages.

In addition, the mRNA levels of EGR1 were also increased in the infected macrophages compared to uninfected cells (**Fig. EV2E in the revised manuscript**).

These results reinforce our observations in the THP-1 model and confirm that the induction of sVEGFR1 and EGR1 also occurs in primary human macrophages. We updated the description in the results section (**Lines 162-166 and Lines 239-241**) and the figure legends (**Lines 1049-1058**). The changes in the revised manuscript were highlighted in yellow.

Figure EV2

Figure EV2. SFTSV infection up-regulated sVEGFR1 and EGR1 expression in primary macrophages and BMDM.

(**A-B, D-E**) PBMCs were isolated by centrifugation over a Ficoll-Paque density gradient. CD14⁺ monocytes were further purified by positive selection using CD14 microbeads. Monocytes were differentiated over 7 days into macrophages in RPMI-1640 with 10% heat-inactivated FBS in the

presence of 50 ng/ml recombinant human M-CSF. Macrophages were washed twice with serum-free RPMI and then infected with SFTSV (MOI = 1) for 24 h in RPMI-1640 with 10% heat-inactivated FBS. SFTSV viral RNA (**A**) and sVEGFR1 mRNA (**B**), total VEGFR1 mRNA (**D**) and EGR1 mRNA (**E**) were measured by qPCR. n = 3 biological replicates. Statistical significance was determined by two-tailed unpaired *Student t*-test. (A: *** $P < 0.0001$; B: *** $P = 0.0002$; D: * $P = 0.0198$; E: ** $P = 0.0046$)

(C) The secreted sVEGFR1 in cell supernatant was measured by capture ELISA. n = 3 biological replicates. Statistical significance was determined by two-tailed unpaired *Student t*-test. (** $P = 0.0026$)

(**F-H**) Bone marrow-derived macrophages (BMDMs) were obtained by in vitro differentiation of the primary femur and tibia BM cells. Briefly, femurs and tibiae from wild type mice were dissected, cleaned, disinfected in 70% ethanol, and washed with fully supplemented RPMI-1640 medium. After lysing erythrocytes, the BM cells were then cultured in a Petri dish (at a density of 4×10^6 cells/10 cm dish) supplemented with 30% L929 cell-conditioned medium. Cells were matured to macrophage over 5-6 days. Adherent cells were recovered and infected with SFTSV for the experiments. BMDMs were infected with SFTSV for 24h. Viral RNA (**F**), sVEGFR1 (**G**) and EGR1 (**H**) mRNA were measured by qPCR. n = 3 biological replicates. Statistical significance was determined by two-tailed unpaired *Student t*-test. (F: *** $P < 0.0001$; G: * $P = 0.0111$; H: *** $P < 0.0001$)

Data information: Data shown were *Mean* \pm *SD* of three biological replicates. (*, $P < 0.05$; **, $P < 0.01$; ***, $P < 0.001$)

2. Have the authors done a comparison in simulating the age of the animals to mirror real like human infections where age is a factor for disease severity?

A2: We appreciate the reviewer's insightful comment. While aging is recognized as a major risk factor for SFTS severity in humans, several previous studies have suggested that this correlation does not hold in mouse models (Sun *et al*, 2021). Comparative studies using newborn, adult, and aged mice have indicated that, unlike in humans, age does not significantly impact disease progression in mice (Matsuno *et al*, 2017). This suggests that conventional age-associated mouse models may not

accurately recapitulate the age-specific pathogenesis or the worsened outcomes observed in elderly SFTS patients. Ferrets are the only animal model that recapitulated the age-related susceptibility to SFTSV and the associated fatality (Park *et al*, 2022; Park *et al*, 2019). In another study that was not directly related to the current study, we infected young and aged ferrets and observed higher fatality of the aged ferrets, consistent with previously reported (Park *et al.*, 2019). I will conduct respective analysis and will present the results in a separate report (manuscript in preparation).

While we acknowledge the importance of age as a factor in human disease severity, the use of young adult humanized mice in this study was necessary to ensure consistent engraftment and experimental reproducibility.

3. There are available data that depicts WNV severity and hemorrhagic fever and overlapping disease manifestations. Author should do a meta-analysis on these datasets.

A3: We appreciate the reviewer's suggestion regarding a meta-analysis of datasets on West Nile virus (WNV) severity, hemorrhagic fever, and overlapping disease manifestations. While we recognize the scientific value of such a comparative approach, the primary focus of our current study is to elucidate the molecular mechanisms underlying SFTSV-induced vascular dysfunction, with an emphasis on the regulatory role of EGR1 in sVEGFR1 expression. Conducting a comprehensive meta-analysis of WNV-related data would require a dedicated systematic review and analytical framework, which is beyond the scope of the present work.

That said, we fully agree that comparative analyses across viral hemorrhagic fevers could offer valuable insights into shared or distinct pathogenic mechanisms. We will consider pursuing such investigations in future studies.

In support of the clinical relevance of hemorrhagic manifestations in SFTS, we would like to highlight that hemorrhagic symptoms are commonly observed in infected patients. In a large observational cohort study, over one-third of SFTS patients exhibited hemorrhagic symptoms (Li *et al*, 2018). Importantly, the presence of

hemorrhagic manifestations was significantly associated with disease severity and increased mortality, underscoring the critical need to monitor hemorrhagic symptoms throughout the disease course (Wang *et al*, 2022a).

Referee #3:

Comments to the Author

The authors show in this study that SFTSV infection leads to an upregulation of the EGR1-sVEGFR1 axis, with a consequential impairment/reduction of angiogenesis both in vitro and in vivo. sVEGFR1 is upregulated in an EGR1-dependent manner and seems one of the main reasons of vascular leakage and hence haemorrhage in addition to thrombocytopenia in haemorrhagic fever virus infection. By using knock-out models, pathogenesis can be ameliorated as well as depletion of sVEGFR1 or targeting sVEGFR1 via an antibody indicated that the mechanism could be therapeutically exploited. The experiments are comprehensive and overall the conclusions are supported by the data. In general, the study would benefit from confirming key findings in the relevant model of primary human macrophages (-derived from monocytes).

I have the following comments:

A: We would like to thank the reviewer for his/her efforts in pointing out the shortcomings of the manuscript. We have considered all comments, performed additional experiments and revised the manuscript to address the reviewer's concerns.

1. It is of interest that sVEGFR1 is more abundantly upregulated in M2-like macrophages and that the EGR1-KO attenuated viral replication, hinting that either EGR1 or its downstream effectors are potentially viral dependency factors. However the enthusiasm related to these results is largely limited by the usage of differentiated THP1 cells for this experiment. It is straightforward to differentiate M1/M2 macrophages from primary human monocytes, so I wonder why the authors, even though corroborating their results with primary cells throughout, fail to do so here.

A1: The reviewer's suggestions are well taken. We conducted additionally experiments using primary human monocyte-derived macrophages to validate our findings. The data were presented below and the experimental descriptions were as follows:

CD14⁺ monocytes were isolated from PBMCs via positive selection using CD14 microbeads and subsequently differentiated into macrophages over 7 days in the presence of recombinant human M-CSF. These macrophages were then differentiated into M1 macrophages by culture in the presence of IFN- γ (20 ng/ml) and LPS (20 ng/ml) for 24 h, or were differentiated into M2 macrophages by culture in the presence of interleukin 4 (20 ng/ml) and interleukin 13 (20 ng/ml) for 72 h. sVEGFR1 expression in macrophages was measured and the results were combined with the data of **Fig. S3C-D** of the original manuscript and now labelled as **Fig. EV3C-F** and the original **3C-D** became **3A-B** in the revised manuscript. Total VEGFR1 and sVEGFR1 mRNAs were measured by qPCR and the results showed that the mRNA levels of total VEGFR1 and sVEGFR1 were increased in M2 macrophage (**Fig. EV3C-D in the revised manuscript**). To further assess the secretion of sVEGFR1 in the supernatants, we performed an ELISA and demonstrated that M2 macrophages produced significantly higher sVEGFR1 in the supernatants as compared with that of M1 macrophages (**Fig. EV3E in the revised manuscript**). Collectively, these results suggested that sVEGFR1 was mainly produced by M2 macrophages and consistent with observations in THP-1 cells.

In addition, the mRNA levels of EGR1 were also increased in M2 macrophages compared to M1 macrophages (**Fig. EV3F in the revised manuscript**). We updated the description in the results section (**Lines 166-173**) and the figure legends (**Lines 1077-1086**). The changes in the revised manuscript were highlighted in yellow.

Figure EV3

Figure EV3. sVEGFR1 mainly produced by M2 macrophages.

(A-B) THP-1 cells were induced to differentiate into M1 macrophage-like cells by 24 h incubation with 150 nM PMA, followed by IFN- γ (20 ng/ml) and LPS (10 pg/ml), or THP-1 cells were induced to differentiate into M2 macrophage-like cells by 24 h incubation with 150 nM PMA, followed by interleukin 4 (20 ng/ml) and interleukin 13 (20 ng/ml) for 72 h. Total VEGFR1 (A) and sVEGFR1 mRNAs (B) were measured by qPCR. $n = 3$ biological replicates. Statistical significance was determined by two-tailed unpaired *Student t*-test. (A: *** $P < 0.0001$; B: *** $P < 0.0001$)

(C-D, F) PBMCs were isolated by centrifugation over a Ficoll-Paque density gradient. CD14⁺ monocytes were further purified by positive selection using CD14 microbeads. Monocytes were differentiated over 7 days into macrophages in RPMI-1640 with 10% heat-inactivated FBS in the presence of 50 ng/ml recombinant human M-CSF. These macrophages were then differentiated into M1 macrophages by culture in the presence of IFN- γ (20 ng/ml) and LPS (20 ng/ml) for 24 h, or were differentiated into M2 macrophages by culture in the presence of interleukin 4 (20 ng/ml) and interleukin 13 (20 ng/ml) for 72 h. Total VEGFR1 (C), sVEGFR1 (D) and EGR1 mRNA (F)

were measured by qPCR. $n = 3$ biological replicates. Statistical significance was determined by two-tailed unpaired *Student t*-test. (C: $**P = 0.0036$; D: $***P < 0.0001$; F: $**P = 0.0018$)

(E) The secreted sVEGFR1 in cell supernatant was measured by capture ELISA. $n = 3$ biological replicates. Statistical significance was determined by two-tailed unpaired *Student t*-test. ($**P = 0.0011$)

Data information: Data shown were *Mean \pm SD* of three biological replicates. ($**$, $P < 0.01$; $***$, $P < 0.001$)

2. In Fig. 1A, the qPCR was done up to 60 h.p.t. and shows a nearly 300x increase in sVEGFR1 while the latest WB timepoint is at 48 h.p.t.. Do the authors have any reason for not processing the 60 h.p.t. for WB as well?

A2: We thank the reviewer for the thoughtful comment. The 60 h.p.t. timepoint for Western blot was actually performed; however, it was not included in the figure as the protein levels began to plateau after 48 h.p.t., showing no substantial increase. Notably, sVEGFR1 levels in the supernatant continued to rise, indicating active secretion of the protein. To provide a more comprehensive view of the expression dynamics, we have now included the full time-course Western blot analysis including the 60 h.p.t. timepoint in **Figure 1B** in the revised manuscript, as shown below:

Figure 1

3. Fig.1: How do the authors justify that they had more vRNA at 36 h.p.i. (see panel F) compared to a later time point (see panel G)?

A3: We appreciate the reviewer's thoughtful comment. The apparent discrepancy in vRNA levels between 36 h.p.i. and the later time point in Figure 1G is likely due to differences in experimental setup. Because we analyzed a broad range of MOI conditions across multiple samples, the assays were conducted using different batches of 96-well plates and qPCR instruments. These technical variations may have introduced minor inconsistencies in relative quantification, although the overall trends remained internally consistent.

To address this concern, we have now repeated the experiment using the same batch of 384-well plates under uniform assay conditions. The new results, presented in the revised **Appendix Figure S9**, confirm that viral RNA levels increase progressively over time. These updated data have been incorporated into the revised manuscript (**Figure 1D-H**).

Appendix Figure S9

Appendix Figure S9.

(A) THP-1 cells were infected with SFTSV for 12, 24, 36, 48 and 60 h at various MOIs (MOI = 0.1, 1, and 10). Viral RNA was determined by qPCR. $n = 3$ biological replicates. Data shown were *Mean \pm SD* of three biological replicates.

Figure 1

4. Line 141: Again here and throughout, be precise in description of the model used; THP1 cells are not monocytes/macrophages; when differentiated they are macrophage-like and without differentiation a monocytic cell line.

A4: We appreciate reviewer's suggestions. We have corrected the description of THP-1 cells as the model used in the revised manuscript, and the changes are highlighted in yellow (**Line 116, Line 154 and Lines 156-157**).

5. Figure 3: It is unclear to me how was this quantified. Did the authors quantify the number of tubes or the area?

A5: Thanks for your comments. For the tube formation assay, quantification was performed using the Angiogenesis Analyzer plug-in of ImageJ software. Specifically, total tube length was measured to assess angiogenesis. For the aortic ring assay, the number of micro-vessels emerging from the aortic rings were quantified, also using ImageJ.

We have presented the respective details in the methods section and highlighted the changes in yellow in the revised manuscript (**Line 594 and Lines 603-604**).

6. Line 183: "significant enhancement of tube formation" This is not shown in Fig. 3D, as the only quantification plotted here is the normalized comparison between healthy control group and the one with the added sVEGFR1.

A6: Thanks for pointing out the error. We have revised the manuscript accordingly, with the changes highlighted in yellow at **Lines 214-219**, as follows:

Meanwhile, the treatment of HUVECs with sera from healthy individuals supported normal tube formation, which was significantly reduced by treatment with recombinant sVEGFR1 protein (**Fig. 3D**). Conversely, the treatment of HUVECs with sera from SFTS patients markedly impaired tube formation, which could be partially restored by treatment with recombinant VEGFA protein.

7. Figure 4: It is important to know which other genes were significantly upregulated, as there are clearly some that are more upregulated than EGR1 in the dot plot.

A7: The reviewer's suggestions are well taken. To provide a more comprehensive view, we have now shown the 81 genes related to angiogenesis among the 422 genes with significant differences in expression (based on an adjusted P value ≤ 0.05) after SFTSV infection in the **Figure 4A** in the revised manuscript, as shown below:

Figure 4

These gene lists have also been included in the **Appendix Table S2** for reference.

Indeed, our transcriptomic analysis revealed multiple genes that were significantly upregulated upon SFTSV infection, and some exhibited even greater changes than EGR1. Nevertheless, we focused on EGR1 due to its known biological relevance in viral pathogenesis and regulation of host responses. Our previous findings have demonstrated the functional importance of EGR1 during SFTSV infection (Jiang *et al*, 2024). Moreover, EGR1 directly regulates the transcription of VEGFR1 by binding to its promoter region (Vidal *et al*, 2000). This regulatory relationship is central to the mechanistic pathway investigated in our study, which links SFTSV infection to vascular dysfunction through EGR1-mediated induction of sVEGFR1.

8. Fig.4: Is EGR1 also upregulated in the serum derived from infected patients?

A8: Thanks for your comments. EGR1 is a nuclear transcription factor and is not secreted into the circulation (Chen *et al*, 2011; Woodson & Kehn-Hall, 2022);

therefore, it cannot be directly measured in serum. In our study, EGR1 expression was assessed at the mRNA and cellular protein level in relevant cell lines, not in serum. However, we did observe upregulation of EGR1 in PBMCs from infected patients, which may contribute to the downstream effects reflected in serum factors such as sVEGFR1. We have provided the data in **Figure EV4**, and we also updated the description in the results section (**Lines 241-243**) and the figure legends (**Lines 1089-1094**). The changes in the revised manuscript were highlighted in yellow.

Images were presented below and the experimental descriptions were as follows: PBMCs were isolated from 10 SFTS patients and 10 healthy individuals as controls. EGR1 expression in PBMCs among SFTS patients were measured by qPCR and the results showed that the mRNA level of EGR1 were increased in PBMCs from infected patients.

Figure EV4. SFTSV infection up-regulated EGR1 expression in SFTS patients.

(A) PBMCs were isolated from 10 SFTS patients and 10 healthy individuals as controls. EGR1 expression in PBMCs among SFTS patients were measured by qPCR. $n = 10$ biological replicates. Statistical significance was determined by two-tailed unpaired *Student t*-test. (** $P = 0.0085$)

Data information: Data shown were *Mean ± SD* of ten biological replicates with each data point representing a biological experiment. (**, $P < 0.01$)

9. Fig.5: It would make more sense to swap panels A and B, to be consistent with all previous figures, showing first the mRNA and then the protein data.

A9: We appreciate reviewer's suggestions. We have swapped panels **A** and **B**, as well as panels **D** and **E** in **Figure 5**, to be consistent with all the previous figures. We also updated the description in the results section and the figure legends. The changes in revised manuscript were highlighted in yellow.

10. Fig.5g: Why are the statistical tests performed between siRNA 1 and siRNA3, instead of the ctrl siRNA?

A10: Apology for the ambiguity. The statistical tests in panel G of Figure 5 were performed between siNC and three different siEGR1 treatments to directly assess the efficacy of EGR1 knockdown on sVEGFR1 expression. We have corrected them in **Figure 5G** in the revised manuscript as following:

11. Fig.7e: Is the AUC significantly different between the Ctrl and the EGR1-KO mice?

A11: We appreciate the reviewer's comment. We would like to clarify that Fig. 7e presents the relative expression of sVEGFR1 mRNA and SFTSV vRNA in WT and EGR1-KO mice, and does not include an AUC (area under the curve) analysis.

In general, AUC analysis requires a sufficiently large sample size to ensure statistical robustness. However, due to the extremely low homozygote ratio of the new born in EGR1 KO mice, large-scale breeding is challenging (Lee *et al*, 1996). Specifically, homozygous EGR1 KO mice were generated by crossing heterozygous parents, yielding a low frequency of homozygous offspring (11 out of 142 pups,

approximately 7.7%; [IMPC, <https://www.mousephenotype.org/>]). This limitation prevented us from performing a statistically meaningful AUC analysis in the mouse model.

Nonetheless, to explore the diagnostic potential of sVEGFR1, we performed an AUC analysis using patient serum data from Fig. 2E. The results demonstrated that the AUC for sVEGFR1 was 0.849 for mild cases and 0.987 for severe cases, indicating strong predictive value in the clinical context.

12. Lines 273-275: also suggests that EGR1 KO mice are less susceptible to SFTSV. Does the viral load correlate with the lower NP levels?

A12: We appreciate reviewer's insightful comments. qPCR analysis confirmed that SFTSV vRNA levels in the blood were significantly lower in EGR1 KO mice compared to WT mice, supporting that the viral load correlates with the reduced NP levels observed in EGR1 KO mice. This suggested that EGR1 deficiency might impair viral replication or dissemination, thereby contributing to decreased disease severity.

We have revised the manuscript accordingly, with the changes highlighted in yellow at **Lines 319-323**, as follows:

Collectively, these results demonstrate that sVEGFR1 expression were reduced in EGR1 KO mice during SFTSV infection, alleviating angiogenesis dysfunction. Notably, EGR1 KO mice also exhibited lower levels of SFTSV NP protein and viral RNA, suggesting that EGR1 KO mice were less susceptible to SFTSV.

13. Lines 306-307: the authors should add that this was observed in THP-1 cells. At first glance, I thought this data came from BMDMs.

A13: We appreciate reviewer's suggestions. We have modified this sentence according to the suggestions and highlighted the changes in yellow in **Line 144** as following:

Meanwhile, qPCR and flow cytometry analysis showed that total VEGFR1 mRNA and protein expression on the cell membrane were significantly increased in THP-1

cells (**Fig. EV1A-B**), suggesting that SFTSV infection upregulated *de novo* production of sVEGFR1.

14. Lines 317-318: would be important to confirm in primary human monocyte-derived-macrophages

A14: The reviewer's suggestions are well taken. We conducted additionally experiments using primary human monocyte-derived macrophages to validate our findings. The images were presented below and the experimental descriptions were as follows:

CD14⁺ monocytes were isolated from PBMCs via positive selection using CD14 microbeads and subsequently differentiated into macrophages over 7 days in the presence of recombinant human M-CSF. These macrophages were then infected with SFTSV for 24 h. sVEGFR1 expression in macrophages were measured and the results were combined with the data of **Fig. S1** of the original manuscript and now labelled as **Fig. EV2A-E** and the original **1A-C** became **2F-H** in the revised manuscript. The results showed that the mRNA level of sVEGFR1 and total VEGFR1 were increased in the infected macrophages (**Fig. EV2A-B, D in the revised manuscript**). To further assess the capability of SFTSV inducing sVEGFR1 secretion, we performed an ELISA and demonstrated that SFTSV-infected macrophages produced significantly higher sVEGFR1 in the supernatants as compared with that of the uninfected macrophages (**Fig. EV2C in the revised manuscript**). Collectively, these results unequivocally confirmed that SFTSV infection triggered the release of sVEGFR1 from primary macrophages.

In addition, the mRNA levels of EGR1 were also increased in the infected macrophages compared to uninfected cells (**Fig. EV2E in the revised manuscript**). These results reinforce our observations in the THP-1 model and confirm that the induction of sVEGFR1 and EGR1 also occurs in primary human macrophages. We updated the description in the results section (**Lines 162-166 and Lines 239-241**) and the figure legends (**Lines 1049-1058**). The changes in the revised manuscript were highlighted in yellow.

Figure EV2

Figure EV2. SFTSV infection up-regulated sVEGFR1 and EGR1 expression in primary macrophages and BMDM.

(A-B, D-E) PBMCs were isolated by centrifugation over a Ficoll-Paque density gradient. CD14⁺ monocytes were further purified by positive selection using CD14 microbeads. Monocytes were differentiated over 7 days into macrophages in RPMI-1640 with 10% heat-inactivated FBS in the presence of 50 ng/ml recombinant human M-CSF. Macrophages were washed twice with serum-free RPMI and then infected with SFTSV (MOI = 1) for 24 h in RPMI-1640 with 10% heat-inactivated FBS. SFTSV viral RNA (A) and sVEGFR1 mRNA (B), total VEGFR1 mRNA (D) and EGR1 mRNA (E) were measured by qPCR. n = 3 biological replicates. Statistical

significance was determined by two-tailed unpaired *Student t*-test. (A: *** $P < 0.0001$; B: *** $P = 0.0002$; D: * $P = 0.0198$; E: ** $P = 0.0046$)

(C) The secreted sVEGFR1 in cell supernatant was measured by capture ELISA. $n = 3$ biological replicates. Statistical significance was determined by two-tailed unpaired *Student t*-test. (** $P = 0.0026$)

(F-H) Bone marrow-derived macrophages (BMDMs) were obtained by in vitro differentiation of the primary femur and tibia BM cells. Briefly, femurs and tibiae from wild type mice were dissected, cleaned, disinfected in 70% ethanol, and washed with fully supplemented RPMI-1640 medium. After lysing erythrocytes, the BM cells were then cultured in a Petri dish (at a density of 4×10^6 cells/10 cm dish) supplemented with 30% L929 cell-conditioned medium. Cells were matured to macrophage over 5-6 days. Adherent cells were recovered and infected with SFTSV for the experiments. BMDMs were infected with SFTSV for 24h. Viral RNA (F), sVEGFR1 (G) and EGR1 (H) mRNA were measured by qPCR. $n = 3$ biological replicates. Statistical significance was determined by two-tailed unpaired *Student t*-test. (F: *** $P < 0.0001$; G: * $P = 0.0111$; H: *** $P < 0.0001$)

Data information: Data shown were *Mean* \pm *SD* of three biological replicates. (*, $P < 0.05$; **, $P < 0.01$; ***, $P < 0.001$)

References:

- Chen J, Liu MY, Parish CR, Chong BH, Khachigian L (2011) Nuclear import of early growth response-1 involves importin-7 and the novel nuclear localization signal serine-proline-serine. *Int J Biochem Cell Biol* 43: 905-912
- Ji M, Hu J, Zhang D, Huang B, Xu S, Jiang N, Chen Y, Wang Y, Wu X, Wu Z (2024) Inhibition of SFTSV replication in humanized mice by a subcutaneously administered anti-PD1 nanobody. *EMBO Mol Med* 16: 575-595
- Jiang N, He Y, Wu J, You Q, Zhang R, Cheng M, Liu B, Cai Y, Lyu R, Wu Z (2024) 6-Thioguanine inhibits severe fever with thrombocytopenia syndrome virus through suppression of EGR1. *Antiviral Res* 227: 105916
- Lee SL, Sadovsky Y, Swirnoff AH, Polish JA, Goda P, Gavrilina G, Milbrandt J (1996) Luteinizing hormone deficiency and female infertility in mice lacking the transcription factor NGFI-A (Egr-1). *Science* 273: 1219-1221

Li H, Lu QB, Xing B, Zhang SF, Liu K, Du J, Li XK, Cui N, Yang ZD, Wang LY *et al* (2018) Epidemiological and clinical features of laboratory-diagnosed severe fever with thrombocytopenia syndrome in China, 2011-17: a prospective observational study. *Lancet Infect Dis* 18: 1127-1137

Li XK, Yang ZD, Du J, Xing B, Cui N, Zhang PH, Li H, Zhang XA, Lu QB, Liu W (2017) Endothelial activation and dysfunction in severe fever with thrombocytopenia syndrome. *PLoS Negl Trop Dis* 11: e0005746

Li Y, Li H, Wang H, Pan H, Zhao H, Jin H, Jie S (2019) The proportion, origin and pro-inflammation roles of low density neutrophils in SFTS disease. *BMC Infect Dis* 19: 109

Matsuno K, Orba Y, Maede-White K, Scott D, Feldmann F, Liang M, Ebihara H (2017) Animal Models of Emerging Tick-Borne Phleboviruses: Determining Target Cells in a Lethal Model of SFTSV Infection. *Front Microbiol* 8: 104

Park SJ, Kim YI, Casel MA, Kim EH, Kim SM, Yu KM, Rollon R, Jang SG, Jeong HW, Choi YK (2022) Infection Route Impacts the Pathogenesis of Severe Fever with Thrombocytopenia Syndrome Virus in Ferrets. *Viruses* 14

Park SJ, Kim YI, Park A, Kwon HI, Kim EH, Si YJ, Song MS, Lee CH, Jung K, Shin WJ *et al* (2019) Ferret animal model of severe fever with thrombocytopenia syndrome phlebovirus for human lethal infection and pathogenesis. *Nat Microbiol* 4: 438-446

Sun J, Min YQ, Li Y, Sun X, Deng F, Wang H, Ning YJ (2021) Animal Model of Severe Fever With Thrombocytopenia Syndrome Virus Infection. *Front Microbiol* 12: 797189

Vidal F, Aragonés J, Alfranca A, de Landazuri MO (2000) Up-regulation of vascular endothelial growth factor receptor Flt-1 after endothelial denudation: role of transcription factor Egr-1. *Blood* 95: 3387-3395

Wang Y, Song Z, Wei X, Yuan H, Xu X, Liang H, Wen H (2022a) Clinical laboratory parameters and fatality of Severe fever with thrombocytopenia syndrome patients: A systematic review and meta-analysis. *PLoS Negl Trop Dis* 16: e0010489

Wang YN, Zhang YF, Peng XF, Ge HH, Wang G, Ding H, Li Y, Li S, Zhang LY, Zhang JT *et al* (2022b) Mast Cell-Derived Proteases Induce Endothelial Permeability and Vascular Damage in Severe Fever with Thrombocytopenia Syndrome. *Microbiol Spectr* 10: e0129422

Woodson CM, Kehn-Hall K (2022) Examining the role of EGR1 during viral infections. *Front Microbiol* 13: 1020220

Xu S, Jiang N, Nawaz W, Liu B, Zhang F, Liu Y, Wu X, Wu Z (2021) Infection of humanized mice with a novel phlebovirus presented pathogenic features of severe fever with thrombocytopenia syndrome. *PLoS Pathog* 17: e1009587

Dear Dr. Wu,

Thank you for the submission of your revised manuscript to our editorial offices. I have now received the report from one of the three referees that I asked to re-evaluate the study, you will find below. Referees #1 and #2 were completely unresponsive to my invitations but going through your p-b-p-response, I consider their concerns and requests as adequately addressed. As you will see, referee #3 now supports the publication of your study. S/he has some final suggestion to improve the manuscript, I ask you to address in a final revised manuscript. Please also provide a final p-b-p-response to these points and the editorial requests below.

Editorial requests:

- Please provide a more comprehensive and compact title.
- Please provide the abstract written in present tense throughout.
- Please also provide the subtitles of the results part and the figure titles written in present tense.
- Please check again that the number "n" for how many independent experiments were performed, their nature (biological versus technical replicates), the bars and error bars (e.g. SEM, SD) and the test used to calculate p-values is indicated in the respective figure legends. Please also check that all the p-values are explained in the legend, and that these fit to those shown in the figure. Please provide statistical testing where applicable. Please avoid the phrase 'independent experiment' but clearly state if these were biological or technical replicates. Please also indicate (e.g. with n.s.) if testing was performed, but the differences are not significant. In case n=2, please show the data as separate datapoints without error bars and statistics. See also:
<http://www.embopress.org/page/journal/14693178/authorguide#statisticalanalysis>

If $n < 5$, please show single datapoints for diagrams. Moreover:

- Please note that the legend for figure EV2 is not provided in the sequential manner (legend for figure D-E is provided before legend of figure C). This needs to be rectified.
- Please note that the legend for figure EV3 is not provided in the sequential manner (legend for figure F is provided before legend of figure E). This needs to be rectified.
- Please note that the exact p values are not provided in the legends of figures 2D, E; 3A, B; 5A, D, F, G; 7J; EV1 A, C, D; EV2 F, H; EV3 D,

- Please indicate the statistical test used for data analysis in the legend of figure 4B
- Please note that the red arrow heads are not defined in the legends of figures 3A, B, C, D. This needs to be rectified.

- Please make sure that all figure panels (main, EV and Appendix figures) are called out separately and sequentially. Presently, there seem to be no callout for Appendix Figures S5-S9 and Appendix Table S2. Please check.

- Please add scale bars of similar style and thickness to microscopic images, using clearly visible black or white bars (depending on the background). Please place these in the lower right corner of the images themselves. Please do not write on or near the bars in the image but define the size in the respective figure legend. Presently, many scalebars are too thin and hard to see, and some have text nearby (e.g. in 6C and 6D). Please check.

- Please remove now the referee token from the Data Availability Section (DAS) and make sure that the dataset is public latest upon online publication of the manuscript.

- Please move the primer table (Appendix Table S1) to the Reagents & Tools Table and remove it from the Appendix. Please name Appendix Table S2 'Appendix Table S1' and adjust any callouts.

- Please also add the primer information in the section 'SFTSV infection of EGR1 KO mice' of the methods to the Reagents & Tools Table and remove the primers information from this section. Please add callouts to the Reagents & Tools Table.

- During our analysis of the figure integrity, we noted a potential vertical splice in the blots shown in Figure 4F, in particular for GAPDH, which is also present in the source data. Please check. If the blot(s) has(have) been cut, please indicate this by a vertical black line and add explanations to the legend.

- It seems in Figures 4G, 6C and 6D magnifications are shown. Please indicate where these come from by boxing the magnified area in the image with the lower magnification.

- In Figure 6D three of the small cells contain repeated patterns (see attached screenshots). Please clarify what these are.

- In Appendix Figure S4A both blots are heavily overcontrasted. Please provide a revised figure with the original captured and unmodified image.

- Please note that corresponding authors are required to supply an ORCID ID upon submission of a revised manuscript. Please do this for co-corresponding author Zhu. Please find instructions on how to link the ORCID ID to the account in our manuscript tracking system in our Author guidelines:
<http://www.embopress.org/page/journal/14693178/authorguide#authorshipguidelines>

In addition, I would need from you uploaded separately:

I look forward to seeing a new revised version of your manuscript as soon as possible.

Please let me know if you have questions regarding the revision.

Best,

Referee #3:

The authors were very responsive to mine and the other referee comments. They confirmed key findings with primary human monocyte derived macrophages or BMDM from mice.

I wonder why they do not include these important results in the main body of the manuscript but present them in the extended version only.

Furthermore, details on the macrophage generation/differentiation etc are missing in the methods section but explained in the figure legends, which should be avoided.

Apart from that, the conclusions are now firmly backed up by the data.

Dear Dr. Breiling,

Thank you for giving us the opportunity to improve our manuscript further. We appreciate the reviewers for recognizing the scientific merits of our findings and supporting the publication of our manuscript in *EMBO Reports*. We addressed the reviewer's remaining concerns point-by-point and carefully revised our manuscript based on the reviewer's suggestions. We also carefully formatted the final version of our manuscript to ensure that it complies with all the editorial requirements. We believe that this revision has significantly improved the quality of the manuscript. Thank you again for all your kind support.

Best regards,

Allen Z. Wu, Ph.D.
Professor and Director
Center for Public Health Research
Medical School, Nanjing University
Professor and Associate Director
State Key Laboratory of Analytical Chemistry for Life Science
Nanjing University
22 Hankou Road, Nanjing, Jiangsu 210093. China

(The comments of the editor and reviewers are highlighted in blue, which are followed by our responses.)

Editorial requests:

- Please provide a more comprehensive and compact title.

Response: We have done so accordingly.

- Please provide the abstract written in present tense throughout.

Response: We have provided the abstracted written in present tense throughout.

- Please also provide the subtitles of the results part and the figure titles written in present tense.

Response: We have provided the subtitles of the results part and the figure titles written in present tense.

- Please check again that the number "n" for how many independent experiments were performed, their nature (biological versus technical replicates), the bars and error bars (e.g. SEM, SD) and the test used to calculate p-values is indicated in the respective figure legends. Please also check that all the p-values are explained in the legend, and that these fit to those shown in the figure. Please provide statistical testing where applicable. Please avoid the phrase 'independent experiment' but clearly state if these were biological or technical replicates. Please also indicate (e.g. with n.s.) if testing was performed, but the differences are not significant. In case n=2, please show the data as separate datapoints without error bars and statistics. See also:

<http://www.embopress.org/page/journal/14693178/authorguide#statisticalanalysis>

Response: We carefully checked each diagram and made necessary revisions based on the above instructions.

Moreover:

- Please note that the legend for figure EV2 is not provided in the sequential manner (legend for figure D-E is provided before legend of figure C). This needs to be rectified.

Response: Thank you for pointing this out. We have corrected the order of the figure legends for **Figure EV2** to ensure proper sequential arrangement. In the revised manuscript, the data previously shown in **Figure EV2** (revised in response to Referee #3) have been reorganized and partially moved to the main figures as follows:

Figure EV2A-B are now shown as **Figure 1N**,

Figure EV2C is now **Figure 1O**,

Figure EV2D is now **Figure EV1E**,

Figure EV2E is now **Figure 4E**,

Figure EV2F-G are now **Figure 1P**,

Figure EV2H is now **Figure 4F**.

We also updated the descriptions in the results and the figure legends.

- Please note that the legend for figure EV3 is not provided in the sequential manner (legend for figure F is provided before legend of figure E). This needs to be rectified.

Response: Thank you for pointing this out. We have corrected the order of the figure legends for **Figure EV2** (formerly **Figure EV3**) to ensure proper sequential arrangement.

- Please note that the exact p values are not provided in the legends of figures 2D, E; 3A, B; 5A, D, F, G; 7J; EV1 A, C, D; EV2 F, H; EV3 D,

Response: Thank you for pointing this out. The p -values of these figures are all smaller than 0.0001, and the statistical software used (GraphPad Prism) reports these as $p < 0.0001$. Therefore, we are unable to provide actual numerical values.

- Please indicate the statistical test used for data analysis in the legend of figure 4B

Response: Thank you for pointing this out. We have now added the statistical test used for data analysis in the legend of **Figure 4B**.

- Please note that the red arrow heads are not defined in the legends of figures 3A, B, C, D. This needs to be rectified.

Response: Thank you for pointing this out. The information about the red arrow heads has been added to the legends of **Figure 3A, B, C, D**.

- Please make sure that all figure panels (main, EV and Appendix figures) are called out separately and sequentially. Presently, there seem to be no callout for Appendix Figures S5-S9 and Appendix Table S2. Please check.

Response: We carefully reviewed the manuscript to ensure all figure panels are called out in the revised manuscript. The appendix tables and figures have also been carefully reviewed and updated.

As **Appendix Figure S9** contains the same data as **Figure 1D-H**, we have removed it from the Appendix to avoid redundancy.

- Please add scale bars of similar style and thickness to microscopic images, using clearly visible black or white bars (depending on the background). Please place these in the lower right corner of the images themselves. Please do not write on or near the bars in the image but define the size in the respective figure legend. Presently, many scalebars are too thin and hard to see, and some have text nearby (e.g. in 6C and 6D). Please check.

Response: The reviewer's suggestions are well taken. We have updated all scale bars for clarity and consistency, as suggested. The corresponding sizes are indicated in the respective figure legends.

- Please remove now the referee token from the Data Availability Section (DAS) and make sure that the dataset is public latest upon online publication of the manuscript.

Response: The referee token has been removed.

- Please move the primer table (Appendix Table S1) to the Reagents & Tools Table and remove it from the Appendix. Please name Appendix Table S2 'Appendix Table S1' and adjust any callouts.

Response: We have moved the primer table from the Appendix (formerly Appendix Table S1) to the **Reagents & Tools Table**. The former Appendix Table S2 has been renamed **Appendix Table S1**, and all corresponding callouts have been updated accordingly in the revised manuscript.

- Please also add the primer information in the section 'SFTSV infection of EGRI KO mice' of the methods to the Reagents & Tools Table and remove the primers information from this section. Please add callouts to the Reagents & Tools Table.

Response: We have moved the primer information mentioned above from the Methods section to the **Reagents & Tools Table**. Appropriate callouts have been added in the revised manuscript.

- During our analysis of the figure integrity, we noted a potential vertical splice in the blots shown in Figure 4F, in particular for GAPDH, which is also present in the source data. Please check. If the blot(s) has(have) been cut, please indicate this by a vertical black line and add explanations to the legend.

Response: Thank you for your careful assessment. We have re-examined the original blot data for **Figure 4H** (formerly **4F**, revised in response to Referee #3) and confirmed that the GAPDH blot does not contain a vertical splice between lanes and the blot was not cut. The original image shown below was imaged using the Li-Cor Odyssey infrared imaging system and what may appear as a splice is likely due to variation in signal intensity or background.

- It seems in Figures 4G, 6C and 6D magnifications are shown. Please indicate where these come from by boxing the magnified area in the image with the lower magnification.

Response: The reviewer's suggestion is well taken. In **Figures 4I** (formerly **4G**, revised in response to Referee #3), **6C**, and **6D**, we have now indicated the magnified areas by adding boxes in the corresponding lower-magnification images. We also updated the descriptions in the figure legends.

- In Figure 6D three of the small cells contain repeated patterns (see attached screenshots). Please clarify what these are.

Response: Thank you for pointing out this issue. We have carefully re-examined **Figure 6D** and the original image files. The similar patterns observed in three regions of the whole-slide image are not duplicated areas. The image was acquired using the **SQS120P whole-slide scanning system**, which captures tiled images and reconstructs a high-resolution composite through algorithmic stitching. Given that the sample was a Matrigel plug, which can exhibit intrinsic structural regularity, some local pattern similarities may have arisen during the stitching process. We confirm that no image duplication or manipulation occurred.

- In Appendix Figure S4A both blots are heavily overcontrasted. Please provide a revised figure with the original captured and unmodified image.

Response: Thank you for pointing this out. We have replaced the overcontrasted blots in **Appendix Figure S6A** (formerly **Appendix S4A**) with the original, unmodified images captured directly from the imaging system. The revised figure has been included in the updated submission.

Appendix Figure S6

A

- Please note that corresponding authors are required to supply an ORCID ID upon submission of a revised manuscript. Please do this for co-corresponding author Zhu. Please find instructions on how to link the ORCID ID to the account in our manuscript tracking system in our Author guidelines: <http://www.embopress.org/page/journal/14693178/authorguide#authorshipguidelines>

Response: Thank you for the reminder. We have provided the ORCID ID for co-corresponding author Dr. Zhu and linked it to the submission system as instructed.

In addition, I would need from you uploaded separately:

- a short, two-sentence summary of the manuscript (not more than 35 words).

Response: A new file with a two-sentence summary page, highlights, and a schematic abstract page is uploaded to the submission system.

- two to four short (!) bullet points highlighting the key findings of your study (two lines each).

Response: A new file with a two-sentence summary page, highlights, and a schematic abstract page is uploaded to the submission system.

- a schematic summary figure as separate file that provides a sketch of the major findings (not a data image) in jpeg or tiff format (with the exact width of 550 pixels and a height of not more than 400 pixels) that can be used as a visual synopsis on our website.

Response: A new file with a two-sentence summary page, highlights, and a schematic abstract page is uploaded to the submission system.

Referee #3:

The authors were very responsive to mine and the other referee comments. They confirmed key findings with primary human monocyte derived macrophages or BMDM from mice.

Response: We really appreciate the referee' comments and constructive suggestions on our work, which helped improve the manuscript.

I wonder why they do not include these important results in the main body of the manuscript but present them in the extended version only.

Response: We appreciate the reviewer's suggestion. We have moved these data to include them in the main body of the manuscript. These results are now presented in **Figure 1N-P** and **Figure 4E-F** (with the original 4E-G now being relabeled as **4G-I**) in the revised manuscript. We also updated the descriptions in the results and the figure legends accordingly.

Furthermore, details on the macrophage generation/differentiation etc are missing in the methods section but explained in the figure legends, which should be avoided.

Response: Thank you for pointing this out. We have now moved all details on the macrophage generation/differentiation from the figure legends to the methods section as suggested.

Apart from that, the conclusions are now firmly backed up by the data.

Response: We sincerely thank the reviewer once again for his/her time and suggestions on our manuscript.

Dr. Zhiwei Wu
Nanjing university
Medical school
22# Hankou Road, Nanjing
Nanjing, jiangsu 210093
China

Dear Dr. Wu,

I am very pleased to accept your manuscript for publication in the next available issue of EMBO reports. Thank you for your contribution to our journal.

Yours sincerely,
